# LGDN: Language-Guided Denoising Network for Video-Language Modeling

**Haoyu Lu**[1,2]  **Mingyu Ding**[3]  **Nanyi Fei**[1,2]  **Yuqi Huo**[4]  **Zhiwu Lu**[1,2,*]

[1]Gaoling School of Artificial Intelligence, Renmin University of China, Beijing, China
[2]Beijing Key Laboratory of Big Data Management and Analysis Methods
[3]The University of Hong Kong, Pokfulam, Hong Kong
[4]JD Corporation, Beijing, China
{lhy1998, luzhiwu}@ruc.edu.cn

## Abstract

Video-language modeling has attracted much attention with the rapid growth of web videos. Most existing methods assume that the video frames and text description are semantically correlated, and focus on video-language modeling at video level. However, this hypothesis often fails for two reasons: (1) With the rich semantics of video contents, it is difficult to cover all frames with a single video-level description; (2) A raw video typically has noisy/meaningless information (e.g., scenery shot, transition or teaser). Although a number of recent works deploy attention mechanism to alleviate this problem, the irrelevant/noisy information still makes it very difficult to address. To overcome such challenge, we thus propose an efficient and effective model, termed Language-Guided Denoising Network (LGDN), for video-language modeling. Different from most existing methods that utilize all extracted video frames, LGDN dynamically filters out the misaligned or redundant frames under the language supervision and obtains only *2–4 salient frames* per video for cross-modal token-level alignment. Extensive experiments on five public datasets show that our LGDN outperforms the state-of-the-arts by large margins. We also provide detailed ablation study to reveal the critical importance of solving the noise issue, in hope of inspiring future video-language work.

## 1  Introduction

Humans are exposed to the world through a variety of sensory organs, such as eyes, ears, and the sense of touch. In the past few years, multi-modal data (e.g., text or video) has grown and accumulated rapidly on the Internet, which brings the increasing demands for video-language understanding. As one of the fundamental topics, video-language modeling is still challenging due to the heterogeneity of the video-text data. More notably, the video-text data is typically noisy (e.g., misaligned or semi-relevant, as shown in Figure 1), leading to intractable video-language modeling.

The dominant paradigm [8, 30, 13, 14, 45] for video-language modeling is to first extract language features and dense video features via off-the-shelf language and vision models (e.g., BERT [7], 3D CNN [48]), and then model the cross-modal representation by defining the objective function (e.g., triplet loss [17]) within a joint semantic space. Although achieving great success, these methods typically densely sample frames from a full sequence of raw video to obtain richer representation and thus cost excessive computation. Since the heavy computation makes it challenging to train the whole network end-to-end, they often achieve sub-optimal performance in video-language modeling. Recently, ClipBERT [25] proposes a sparse sampling strategy to tackle this drawback. Concretely,

---

*The corresponding author.

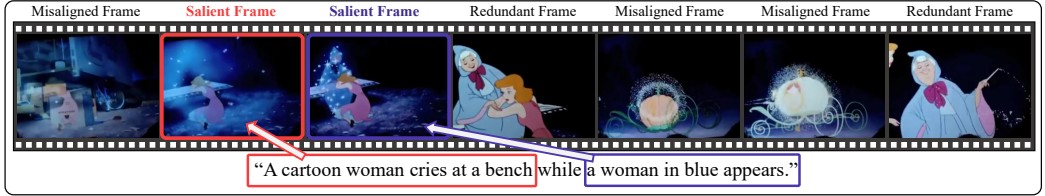

Figure 1: An example of video-text pair where the raw video contains misaligned and redundant video frames given the text description. Instead of aggregating all video frames for token-level alignment, we obtain *only 2–4 salient frames* per video by filtering out the misaligned ones. We find that utilizing 2–4 salient frames is much more effective while enjoying faster speed.

ClipBERT first samples video frames sparsely (8–16 frames per video), and then models the cross-modal alignment at frame-level. This sparse sampling paradigm enables end-to-end training, leading to much better performance. Nevertheless, token-level cross-modal interaction, which has achieved great success in image-text modeling [21, 26], is still not well explored for video-language modeling due to the heavy resource computation (even with 8–16 frames per video). Moreover, both the dominant paradigm and ClipBERT's sparse sampling paradigm assume that video frames and the text description (w.r.t. a video-text pair) are semantically correlated, which is often invalid in practice.

The correlation hypothesis often fails for two reasons: (1) With the rich semantics of video contents, it is hard to cover all frames with a single video-level description; (2) A raw video often has noisy or meaningless information (e.g., scenery shot, transition or teaser). For the dominant paradigm which utilizes densely-sampled frames, though often with self-attention mechanism [43], the irrelevant/noisy information makes it hard to learn high-quality video-language representation. For the sparse sampling paradigm used in ClipBERT that models the cross-modal alignment at frame-level, the misaligned frame-text pairs are wrongly forced to become closer, which inevitably leads to inaccurate cross-modal alignment. Overall, due to this noise issue (see Figure 1), video-language modeling is still challenging. Note that humans also encounter such problem in reality, but seem to be born with the ability to resist noise. That is, everyone can quickly scan through the entire video, easily ignore the noisy frames and focus on the salient ones given the text.

Motivated by this human ability, we propose a **L**anguage-**G**uided **D**enoising **N**etwork termed LGDN to dynamically filter out irrelevant or redundant information under the language supervision for better video-language modeling. Concretely, we devise a **S**alient **F**rame **P**roposal (SFP) mechanism which adopts four strategies to estimate frame-level relevance scores under the language supervision and proposes/selects only salient frames (per video) for precisely video-language modeling. Although the frame embeddings and text embeddings can be (roughly) aligned by introducing a **M**omentum **V**ideo-Level **C**ontrastive **L**earning (MVCL) module, it is vital to precisely establish frame-text alignment for proposing salient frames. Therefore, based on multiple instance learning (MIL), we propose a **M**omentum **F**rame-Level **M**ultiple **S**alient-instance learning (MSL) -**C**ontrastive **L**earning (MFCL) module for video-language modeling at frame-level. Finally, with our SFP mechanism, we propose a **L**anguage-**G**uided **S**alient **F**rame **M**atching (LSFM) module for fine-grained alignment, which adopts a token-aware cross-attention Transformer for cross-modal token-level alignment.

Our main contributions are as follows: (1) We devise a salient frame proposal mechanism that can dynamically filter out irrelevant information under the language supervision, meanwhile maintaining salient information. (2) We propose an end-to-end framework termed LGDN for video-language modeling with cross-modal interaction at three levels: language-guided salient frame matching at token-level, momentum frame-level MSL-contrastive learning, and momentum video-level contrastive learning. (3) We evaluate our LGDN on five public datasets and find that our LGDN outperforms the latest competitors by large margins. We also provide detailed ablation study to reveal the critical importance of solving the noise issue, in hope of inspiring future video-language work.

## 2   Related Work

**Video-Language Modeling.**   Video-language modeling, a fundamental research topic that is beneficial for search engine and video recommendation, has attracted a lot of attention in recent years with the rapid growth of web videos. Previous works have made great efforts to model richer rep-

resentations for video and text modalities and then align the features of the two modalities by the objective function (e.g., triplet loss). One common representative approach [5, 20] is to adopt a Graph Convolution Network (GCN) to extract richer information for video-text retrieval. Another representative approach [30, 13, 14, 52, 29] is to exploit extra experts (e.g., object, motion, speech) for video-language modeling. Recently, ClipBERT [25] proposes a sparse sampling strategy that enables end-to-end training, thus achieving higher performance. Moreover, Frozen in Time [2] also follows a sparse sampling paradigm, and proposes an end-to-end trainable model that is designed to take advantage of both large-scale image and video captioning datasets. However, as illustrated in Figure 1, a raw video typically has noisy/meaningless information, and thus the presence of misaligned frames is inevitable during video-language modeling. Note that most existing methods assume that the video frames and paired text are semantically correlated, without considering the noise phenomenon. Although a self-attention mechanism has been widely applied, the misaligned frames still harm the cross-modal alignment. In this work, we thus propose a salient frame proposal mechanism to effectively (and directly) address this problem.

**Cross-Modal Alignment Objective Functions**  Most previous methods adopt triplet loss as a major objective function for video-language modeling. CGMSCD [14] points out that the triplet loss sometimes leads to a wrong learning direction and thus devises an adaptive margin triplet loss for representation learning. More recent works [41, 12, 19] propose to apply the InfoNCE contrastive loss [47, 38, 6] to enhance representation learning. Particularly, BriVL [18], ALBEF [26] and COTS [32] introduce a momentum mechanism [15] to maintain more negative samples for image-text contrastive learning. Following these state-of-the-art models, we propose momentum video-level contrastive learning for video-text global alignment in this paper. Note that MIL-NCE [35] enhances the InfoNCE loss with multiple-instance learning (MIL) to cope with the misaligned narration descriptions in HowTo100M [36]. In this work, we thus propose momentum frame-level MSL-contrastive learning to assist in addressing the misaligned frame problem.

## 3   Methodology

Figure 2 gives a brief overview of our LGDN framework for video-language modeling, which is composed of four main components: 1) language and vision representation extractors; 2) momentum video-level contrastive learning; 3) momentum frame-level MSL-contrastive learning, and 4) language-guided salient frame matching. In the following, we will describe each component in detail.

### 3.1   Feature Representation

**Vision Representation.**   Given an input video $V$ as a sequence of frames $\{E_i\}_{i=1}^{N}$, where $N$ is the length of the video, we utilize a 2-D vision Transformer (e.g., ViT) as our vision backbone to extract frame-level features $\mathbf{E} = \{\mathbf{E}_1, \mathbf{E}_2, ..., \mathbf{E}_N\}$. Each frame $E_i$ of video $V$ can be represented as $\mathbf{E}_i = [\mathbf{e}_{cls}; \mathbf{e}_1; ...; \mathbf{e}_{k_v-1}] \in R^{k_v \times D_v}$, where $\mathbf{e}_{cls}$ denotes the [CLS] token, $k_v$ denotes the patch sequence length, and $D_v$ denotes the dimension of the patch embeddings. We utilize a fully-connected layer to project the [CLS] token into the frame embedding $\mathbf{f}_i^e$. We then deploy a temporal module $T$ (e.g., a Transformer layer) to aggregate the frame embeddings to obtain the final video embedding:

$$\mathbf{f}^v = T([\mathbf{f}_1^e, \mathbf{f}_2^e, ..., \mathbf{f}_N^e]) = f^v(V), \tag{1}$$

where $f^v$ denotes the entire vision (video) encoder.

**Language Representation.**   Given an input text $L$, we utilize BERT-Base as our language backbone to extract text feature $\mathbf{L}$, which can be represented as $\mathbf{L} = [\mathbf{l}_{cls}; \mathbf{l}_1; ...; \mathbf{l}_{k_l-1}] \in R^{k_l \times D_l}$, where $\mathbf{l}_{cls}$ is the [CLS] token, $k_l$ is the token sequence length, and $D_l$ is the dimension of the token embeddings. We deploy a fully-connected layer to project the [CLS] token into the text embedding $\mathbf{f}^l = f^l(L)$, where $f^l$ is the language encoder.

### 3.2   Momentum Video-Level Contrastive Learning (MVCL) Module

Note that our LGDN is designed to filter out the unmatched/redundant frames for better token-level alignment, without leveraging the temporal information of the videos explicitly. Therefore, we firstly introduce a **M**omentum **V**ideo-**L**evel **C**ontrastive **L**earning (MVCL) module to address this problem.

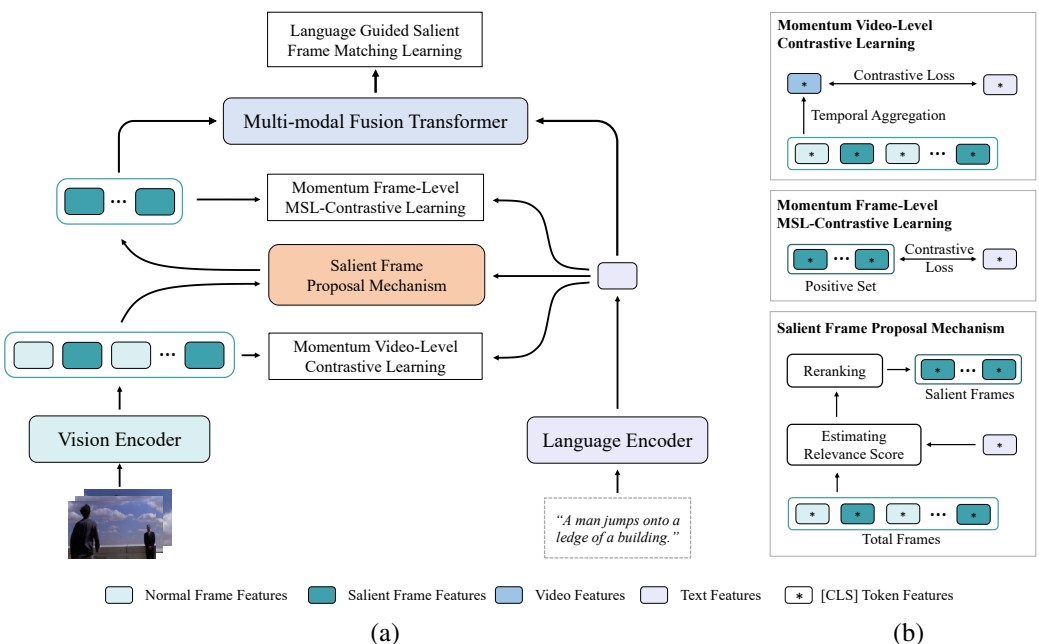

Figure 2: (a) A schematic illustration of the proposed LGDN framework. (b) Details of each component of our LGDN. * denotes that calculation is performed only at the [CLS] token level.

The MVCL module utilizes a temporal module (e.g., Transformer block) to aggregate the frame embeddings to obtain the video embedding. Contrastive learning is then applied for holistic video-text alignment. However, video data takes up large GPU memory and the mini-batch size tends to be small with strict resource, which brings harm to contrastive learning. Inspired by MoCo [15], we introduce the momentum mechanism to maintain massive negative samples in memory bank for contrastive learning. Concretely, We firstly maintain video memory bank $\mathcal{M}^v = \{\hat{\mathbf{q}}_j^v\}_{j=1}^{N_m}$ and text memory bank $\mathcal{M}^l = \{\hat{\mathbf{q}}_j^l\}_{j=1}^{N_m}$ to store video/text features, where $N_m$ denotes the memory bank size and $\hat{\mathbf{q}}_j^v / \hat{\mathbf{q}}_j^l$ denotes the $j$-th stored video/text feature vector. Let $f^v$ (with parameters $\theta^v$) and $\hat{f}^v$ (with parameters $\hat{\theta}^v$) denote vision encoder and vision momentum encoder, respectively. Similarly, let $f^l$ (with parameters $\theta^l$) and $\hat{f}^l$ (with parameters $\hat{\theta}^l$) denote language encoder and language momentum encoder, respectively. The parameters of momentum encoders are updated by:

$$\hat{\theta}^v = m \cdot \hat{\theta}^v + (1-m) \cdot \theta^v, \quad \hat{\theta}^l = m \cdot \hat{\theta}^l + (1-m) \cdot \theta^l, \tag{2}$$

where $m$ is the momentum coefficient hyper-parameter.

The loss function is thus constructed as follow: for each video $V_i$ in mini-batch $\mathcal{B}$, we define the video-to-text contrastive loss between its paired text $L_i$ and all negative samples in the text memory bank $\mathcal{M}^l$, resulting in an InfoNCE loss (with $\tau$ being the temperature hyper-parameter):

$$\mathcal{L}_{\text{V2T}} = -\frac{1}{|\mathcal{B}|} \sum_{(V_i, L_i) \in \mathcal{B}} \log \frac{\exp(\cos(\mathbf{f}_i^v, \hat{\mathbf{f}}_i^l)/\tau)}{\exp(\cos(\mathbf{f}_i^v, \hat{\mathbf{f}}_i^l)/\tau) + \sum_{\hat{\mathbf{q}}_j^l \in \mathcal{M}^l} \exp(\cos(\mathbf{f}_i^v, \hat{\mathbf{q}}_j^l)/\tau)}, \tag{3}$$

where $\hat{\mathbf{f}}_i^l = \hat{f}^v(l_i)$, and the similarity of two features is measured by the cosine similarity. Similarly, given each text description $L_i$ in mini-batch $\mathcal{B}$, we define the text-to-video contrastive loss as:

$$\mathcal{L}_{\text{T2V}} = -\frac{1}{|\mathcal{B}|} \sum_{(V_i, L_i) \in \mathcal{B}} \log \frac{\exp(\cos(\mathbf{f}_i^l, \hat{\mathbf{f}}_i^v)/\tau)}{\exp(\cos(\mathbf{f}_i^l, \hat{\mathbf{f}}_i^v)/\tau) + \sum_{\hat{\mathbf{q}}_j^v \in \mathcal{M}^v} \exp(\cos(\mathbf{f}_i^l, \hat{\mathbf{q}}_j^v)/\tau)}, \tag{4}$$

where $\hat{\mathbf{f}}_i^v = \hat{f}^v(V_i)$. Finally, the objective function for MVCL is defined as follows:

$$\mathcal{L}_{\text{MVCL}} = \mathcal{L}_{\text{V2T}} + \mathcal{L}_{\text{T2V}}. \tag{5}$$

Table 1: Four strategies for estimating relevance scores.

| SimDot | Momentum | CrossMom | Collaborative |
| --- | --- | --- | --- |
| $R(j\|i) = \mathbf{f}_{i,j}^e \cdot \mathbf{f}_i^l$ | $R(j\|i) = \mathbf{f}_{i,j}^e \cdot \mathbf{f}_i^l + \hat{\mathbf{f}}_{i,j}^e \cdot \hat{\mathbf{f}}_i^l$ | $R(j\|i) = \hat{\mathbf{f}}_{i,j}^e \cdot \mathbf{f}_i^l + \mathbf{f}_{i,j}^e \cdot \hat{\mathbf{f}}_i^l$ | $R(j\|i) = (\mathbf{f}_{i,j}^e + \hat{\mathbf{f}}_{i,j}^e) \cdot (\mathbf{f}_i^l + \hat{\mathbf{f}}_i^l)$ |

### 3.3 Salient Frame Proposal (SFP) Mechanism

As shown in Figure 1, video-text data inevitably contains misaligned frame-text pairs. Although an attention mechanism has been applied in Eq. (1), the irrelevant and noisy information would still mislead the cross-modal alignment in our model. To alleviate this problem, we thus propose a **S**alient **F**rame **P**roposal (SFP) mechanism for video-language modeling.

The core idea of our SFP mechanism is to dynamically filter out misaligned or redundant frames and maintain only a few important frames to represent the video well, which are called as salient frames. Formally, for each video-text pair, we first identify the relevance score $R(j|i)$ between the text $L_i$ and the $j$-th frame $E_{i,j}$ of the video $V_i$. Further, we perform language-guided denoising to retain only top-$N_{salient}$ salient frames by filtering out the unmatched/redundant frames from each video.

Since only video-level annotations are provided, we need to estimate the relevance scores $R$ automatically. As shown in Table 1, we introduce four strategies for estimating relevance scores. **(1) SimDot prediction** relies on the output of two separate encoders (i.e., frame encoder $f^e$ and language encoder $f^l$) to model the relevance score $R(j|i)$ by computing the dot product of the frame embedding $\mathbf{f}_{i,j}^e = f^e(E_{i,j})$ and the text embedding $\mathbf{f}_i^l = f^l(L_i)$. However, since video-text data is noisy, only utilizing single-modality encoders may result in incorrect salient frames. **(2) Momentum prediction** improves SimDot prediction by introducing the supervision of momentum encoders (i.e., momentum frame encoder $\hat{f}^e$ and momentum language encoder $\hat{f}^l$), where the momentum frame embedding $\hat{\mathbf{f}}_{i,j}^e = \hat{f}^e(E_{i,j})$ and the momentum text embedding $\hat{\mathbf{f}}_i^l = \hat{f}^l(L_i)$. **(3) CrossMom prediction** considers the frame-text alignment that is directly built on the interaction between one modality encoder and another modality's momentum encoder. **(4) Collaborative prediction** combines Momentum prediction and CrossMom prediction for better performance.

Although the frame embeddings and text embeddings can be (roughly) aligned through applying video-text contrastive learning in Sec. 3.2, it is vital to precisely establish frame-text alignment for proposing/selecting salient frames. To this end, we introduce the MFCL module below.

### 3.4 Momentum Frame-Level MSL-Contrastive Learning (MFCL) Module

To dynamically filter out the unmatched/redundant frames, we propose to adopt frame-level contrastive learning to directly measure the relevance scores $R$ between video frames and paired text. However, video data often contains misaligned frame-text pairs. Simply applying standard NCE-based contrastive learning would force the misaligned frame-text pairs to be pulled closer, which inevitably has negative effect on learning high-quality frame-text representation. Inspired by MIL-NCE [35], we thus propose a **M**omentum **F**rame-Level Multiple Salient-instance Learning (MSL) **C**ontrastive **L**earning (MFCL) module to assist in alleviating the noise problem. The core idea is to use the salient frames filtered by the SFP Mechanism in each video to form a set of positive candidate pairs, instead of considering each positive pair independently. In this work, we suppose that MFCL and SFP have mutual interdependence so that they can bring boost to each other during training.

Similar to MVCL, we additionally maintain a frame-level memory bank $\mathcal{M}^e = \{\hat{\mathbf{q}}_{j'}^e\}_{j'=1}^{N_m*N}$ to store frame features, where $N_m$ is the memory bank size, $N$ is the number of sampled frames per video, and $\hat{\mathbf{q}}_{j'}^e$ is a stored frame feature vector. Given each text description $L_i$ in mini-batch $\mathcal{B}$, we select salient frames filtered by the SFP Mechanism in the paired video $V_i$ to form a set of positive candidate (frame-text) pairs $S_i$ and all frame samples in $\mathcal{M}^e$ to form the negative ones. We then define the text-to-frame contrastive loss as:

$$\mathcal{L}_{\text{T2E}} = -\frac{1}{|\mathcal{B}|} \sum_{(S_i, L_i) \in \mathcal{B}} \log \frac{\sum_{\hat{\mathbf{f}}_{ij}^e \in \hat{\mathbf{f}}_i^s} \exp(\cos(\mathbf{f}_i^l, \hat{\mathbf{f}}_{ij}^e)/\tau)}{\sum_{\hat{\mathbf{f}}_{ij}^e \in \hat{\mathbf{f}}_i^s} \exp(\cos(\mathbf{f}_i^l, \hat{\mathbf{f}}_{ij}^e)/\tau) + \sum_{\hat{\mathbf{q}}_{j'}^e \in \mathcal{M}^e} \exp(\cos(\mathbf{f}_i^l, \hat{\mathbf{q}}_{j'}^e)/\tau)}, \quad (6)$$

where $S_i = \{E_{i,j}\}_{j=1}^N$ is the positive frame set of the video $V_i$, $N$ is the frame sequence length of the video, $\mathbf{f}_i^l = f^l(L_i)$, and $\hat{\mathbf{f}}_i^s = \{\hat{\mathbf{f}}_{ij}^e\}_{j=1}^N = \{\hat{f}^e(E_{i,j})\}_{j=1}^N$.

Similarly, given each positive frame set $S_i$, we define the frame-to-text contrastive loss as:

$$\mathcal{L}_{\text{E2T}} = -\frac{1}{|\mathcal{B}|} \sum_{(S_i, L_i) \in \mathcal{B}} \log \frac{\sum_{\mathbf{f}_{ij}^e \in \mathbf{f}_i^s} \exp(\cos(\mathbf{f}_{ij}^e, \hat{\mathbf{f}}_i^l)/\tau)}{\sum_{\mathbf{f}_{ij}^e \in \mathbf{f}_i^s} \exp(\cos(\mathbf{f}_{ij}^e, \hat{\mathbf{f}}_i^l)/\tau) + \sum_{\mathbf{f}_{ij}^e \in \mathbf{f}_i^s} \sum_{\hat{\mathbf{q}}_{j'}^l \in \mathcal{M}^l} \exp(\cos(\mathbf{f}_{ij}^e, \hat{\mathbf{q}}_{j'}^l)/\tau)},$$
(7)

where $\hat{\mathbf{f}}_i^l = \hat{f}^l(L_i)$ and $\mathbf{f}_i^s = \{\mathbf{f}_{ij}^e\}_{j=1}^N = \{f^e(E_{i,j})\}_{j=1}^N$ (text memory bank $\mathcal{M}^l = \{\hat{\mathbf{q}}_{j'}^l\}_{j'=1}^{N_m}$ is defined in Sec. 3.2). As a result, by combining the text-to-frame and frame-to-text contrastive losses, the objective function for MFCL is given by:

$$\mathcal{L}_{\text{MFCL}} = \mathcal{L}_{\text{E2T}} + \mathcal{L}_{\text{T2E}}.$$
(8)

### 3.5 Language-Guided Salient Frame Matching (LSFM) Module

After obtaining language-guided salient frames, we utilize a multi-modal cross-attention fusion Transformer (see Figure 2) to capture token-level semantic alignment between visual patches and words for better performance (see the design details of this Transformer in the supp. material). Further, we take the [CLS] token embedding outputted by the multi-modal fusion Transformer as the joint representation of a frame-text pair $(V_i, L_i)$, and deploy a fully-connected layer to predict the matched probability, which is similar to the sentence pair classification task in BERT's pre-training phase. The matching loss is defined as:

$$\mathcal{L}_{\text{LSFM}} = -\mathbb{E}_{(\mathbf{E}_{i,j}, \mathbf{L}_i) \sim \mathcal{D}_{salient}} \log P(y_{i,j}|\mathbf{E}_{i,j}, \mathbf{L}_i),$$
(9)

where $\mathbf{E}_{i,j}$ denotes $j$-th frame feature of video $V_i$, $\mathbf{L}_i$ denotes text feature, $\mathcal{D}_{salient}$ is the set of salient frame-text pairs obtained by applying the SFP mechanism to the mini-batch, and $y_{i,j}$ is the ground-truth matching label (0 or 1) of the frame-text pair $(\mathbf{E}_{i,j}, \mathbf{L}_i)$. During inference, we use a mean pooling layer to aggregate all salient frame scores as the video-level prediction score.

Finally, by combining all the proposed modules for video-language modeling at three levels, we train our LGDN model via minimizing the total objective function:

$$\mathcal{L}_{\text{LGDN}} = \mathcal{L}_{\text{MVCL}} + \mathcal{L}_{\text{MFCL}} + \mathcal{L}_{\text{LSFM}}.$$
(10)

## 4 Experiments

### 4.1 Datasets and Settings

**Pre-Training Datasets.** Due to the restricted computing resources, we follow COTS [32] to pre-train our LGDN on the pure image-text datasets. Our pre-training datasets consists of Conceptual Captions [42], SBU [39], VG [23] and MSCOCO [28], which contains 5.2 million image-text pairs. We additionally apply CC12M [3] (about 2 million URLs are now invalid) for better performance, which accumulates 15.2 million image-text pairs in total.

**Downstream Datasets.** We evaluate our proposed LGDN on four public video-text retrieval datasets: MSR-VTT [50], MSVD [4], DiDeMo [16], and VATEX [46]. To further demonstrate the general applicability of our LGDN, we also carry out experiments on a public video-question answering dataset: MSRVTT-QA [49]. We present the details of these downstream datasets as well as the evaluation metrics for downstream tasks in the supp. material.

**Implementation Details.** Following previous work [25], we sample $N = 16$ frames per video: each video is equally split into 16 segments and one frame is randomly sampled from each segment. We empirically set the initial learning rate to 1e-5 and adopt AdamW [31] with a weight decay of 0.02 for 5 epochs. In the warm-up stage (first epoch), the model is trained to optimize Eq. (10) without applying SFP mechanism. We also set the other hyper-parameters uniformly as: salient frame numbers $N_{salient} = 2$, mini-batch size $|\mathcal{B}| = 24$, momentum hyper-parameter $m = 0.99$, temperature $\tau = 0.07$, and queue size $N_m = 9,600$. We adopt pre-trained BERT-Base as language encoder and ViT-Base [9] as vision encoder. More details are given in the supp. material.

**Evaluation Metrics.** We adopt two widely-used metrics in cross-modal retrieval: Recall at K (R@K, K= 1, 5, 10), and Median Rank (MdR) / Mean Rank (MnR). R@K means the percentage of correct matching in the K nearest points, and MdR / MnR measures the median / mean rank of target items in the retrieved ranking list. We also report two additional metrics named 'R@Sum' and 'R@Mean' in our ablation study, which sums/averages all recall metrics for overall evaluation. Following ClipBERT [25], we also report accuracy (Acc) in video-question answering task.

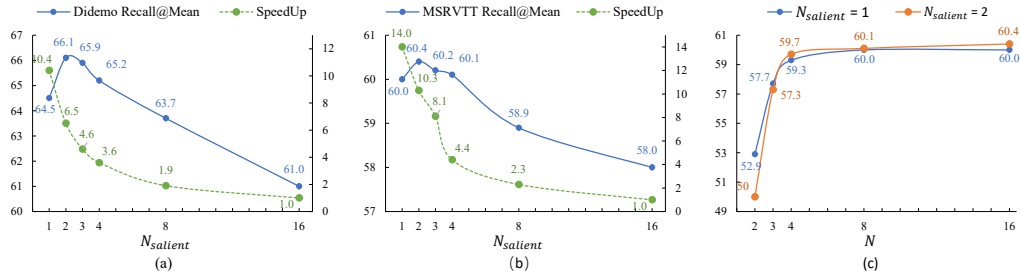

Figure 3: (a-b) Effect of value change of $N_{salient}$. The speedup (green line) is computed w.r.t. the slowest case $N_{salient} = 16$ (w/o SFP). We find that utilizing 2–4 salient frames is much more effective while enjoying faster speed. (c) Effect of value change of $N$ on the MSR-VTT 1k-A test set.

Table 2: Ablation study for our full LGDN model. Retrieval results are reported on the MSR-VTT 1k-A test set. Local: only token-level alignment during inference. Global: only global alignment during inference. Ensemble: ensemble of global and local (token-level) alignment.

| Method | Inference | Text-to-Video Retrieval | | | | Video-to-Text Retrieval | | | | R@SUM |
| | | R@1 | R@5 | R@10 | MdR | R@1 | R@5 | R@10 | MdR | |
| --- | --- | --- | --- | --- | --- | --- | --- | --- | --- | --- |
| $\mathcal{L}_{LSFM}$ (w/o SFP) | Local | 31.4 | 59.8 | 70.3 | 4.0 | 34.9 | 61.9 | 72.9 | 3.0 | 331.2 |
| $\mathcal{L}_{LSFM}$ (w/o SFP) $+\mathcal{L}_{MFCL}$ (w/o MSL) | Local | 32.2 | 58.6 | 70.0 | 3.0 | 34.7 | 61.7 | 73.3 | 3.0 | 330.5 |
| $\mathcal{L}_{LSFM}$ (w/o SFP) $+\mathcal{L}_{MFCL}$ | Local | 33.2 | 60.2 | 71.0 | 3.0 | 34.7 | 62.5 | 73.6 | 3.0 | 335.2 |
| $\mathcal{L}_{LSFM}$ (w/o SFP) $+\mathcal{L}_{MFCL} + \mathcal{L}_{MVCL}$ | Local | 33.0 | 60.4 | 71.2 | 3.0 | 35.6 | 62.2 | 73.7 | 3.0 | 336.1 |
| $\mathcal{L}_{LSFM} +\mathcal{L}_{MFCL} + \mathcal{L}_{MVCL}$ | Local | 35.3 | 65.0 | 75.3 | 3.0 | 36.3 | 65.0 | 76.0 | 3.0 | 352.9 |
| $\mathcal{L}_{LSFM} +\mathcal{L}_{MFCL} + \mathcal{L}_{MVCL}$ | Global | 32.5 | 60.4 | 71.7 | 3.0 | 32.1 | 61.8 | 72.2 | 3.0 | 330.7 |
| $\mathcal{L}_{LSFM} +\mathcal{L}_{MFCL} + \mathcal{L}_{MVCL}$ | Ensemble | **38.9** | **65.7** | **76.5** | **2.0** | **37.9** | **65.4** | **76.0** | **2.0** | **360.4** |

## 4.2 Ablation Study

In this subsection, we conduct comprehensive ablation study to investigate the contributions of different components of our full model. If not specifically indicated, we set $N = 16$ for global alignment and $N_{salient} = 2$ for token-level alignment as the default setting.

**Effect of Value Change of $N_{salient}$ and $N$.** A common perspective for video/video-language understanding is that more frames per video bring better performance. We thus conduct experiments on the frame number used for token-level alignment in Figure 3(a-b). We sample $N = 16$ frames from each video and evaluate different variants that use $N_{salient} \in \{1, 2, 3, 4, 8, 16\}$ frames. Note that when $N_{salient} = 16$, sampling by our SFP degrades to w/o SFP. It can be observed that utilizing only $N_{salient} = \{2, 3, 4\}$ salient frames filtered by our SFP significantly outperforms utilizing all 16 extracted frames meanwhile enjoying the faster speed (see the green lines). This suggests that our SFP mechanism not only selects correct salient frames but also alleviates the noise problem. To investigate the influence of value change of $N$ on our LGDN, we evenly sample $N \in \{2, 3, 4, 8\}$ frames per video and freeze $N_{salient} = \{1, 2\}$ salient frames. The results in Figure 3(c) indicate that more extracted frames per video are beneficial to the token-level alignment in our LGDN model, as it provides larger candidate set for selecting salient frames. Meanwhile, when $N$ becomes larger ($> 4$), the performance tends to converge, further demonstrating the redundancy in the videos.

**Contributions of Each Components.** We further demonstrate the contributions of the three objective functions as well as the salient frame proposal (SFP) mechanism used in our full LGDN model in Table 2. We start with the objective function $\mathcal{L}_{LSFM}$ (w/o SFP), which means only applying matching loss in token-level alignment without using the SFP mechanism. It can be observed that: (1) $\mathcal{L}_{MFCL}$ (and $\mathcal{L}_{MVCL}$) combined with $\mathcal{L}_{LSFM}$ (w/o SFP) can bring improvements, suggesting that global alignment is beneficial to token-level alignment (during the training stage). (2) Simply applying the frame-level alignment may cause negative effect while combing with our MSL design brings better results. This demonstrates that our design of $\mathcal{L}_{MFCL}$ does help alleviate the noise problem. (3) When the SFP mechanism is added (see $\mathcal{L}_{LSFM}$ (w/o SFP) $+\mathcal{L}_{MFCL} + \mathcal{L}_{MVCL}$ vs. $\mathcal{L}_{LSFM} +\mathcal{L}_{MFCL} + \mathcal{L}_{MVCL}$), the performance is significantly improved, which clearly shows the effectiveness of our proposed SFP mechanism. (4) For the same trained full LGDN model, combining the global and token-level alignment during inference can bring further improvements. Note that our full LGDN still achieves the state-of-the-art on MSR-VTT even without considering global alignment during inference.

Table 3: Comparison to the state-of-the-arts for video-text retrieval on MSR-VTT. Extra Expert: methods utilized expert features (e.g., object, motion and OCR features). # PT Pairs: the number of pre-training pairs. $^\dagger$ denotes that our LGDN is additionally pre-trained with CC12M [3].

| Method | Extra Expert | #PT Pairs | Text-to-Video Retrieval | | | | Video-to-Text Retrieval | | | |
|---|---|---|---|---|---|---|---|---|---|---|
| | | | R@1 | R@5 | R@10 | MdR | R@1 | R@5 | R@10 | MdR |
| **Full Split:** | | | | | | | | | | |
| HGR [5] | | - | 9.2 | 26.2 | 36.5 | 24.0 | 15.0 | 36.7 | 48.8 | 11.0 |
| CE [30] | ✓ | - | 10.0 | 29.0 | 41.2 | 16.0 | 15.6 | 40.9 | 55.2 | 8.3 |
| CMGSD [14] | ✓ | >100M | 11.3 | 32.0 | 44.1 | 14.2 | 17.2 | 43.6 | 57.2 | 7.6 |
| T2VLAD [45] | ✓ | >100M | 12.7 | 34.8 | 47.1 | 12.0 | 20.7 | 48.9 | 62.1 | 6.0 |
| **LGDN (ours)** | | 5.2M | **22.9** | **46.0** | **56.8** | **7.0** | **41.8** | **65.2** | **74.6** | **2.0** |
| **LGDN$^\dagger$ (ours)** | | 15.2M | 27.5 | 51.7 | 61.9 | 5.0 | 50.2 | 73.9 | 82.3 | 1.0 |
| **7k-1k Split:** | | | | | | | | | | |
| HERO [27] | | >100M | 16.8 | 43.4 | 57.7 | - | - | - | - | - |
| UniVL [33] | ✓ | >100M | 21.2 | 49.6 | 63.1 | 6.0 | - | - | - | - |
| ClipBERT [25] | | 5.6M | 22.0 | 46.8 | 59.9 | 6.0 | - | - | - | - |
| TACo [52] | ✓ | >100M | 24.8 | 52.1 | 64.5 | 5.0 | - | - | - | - |
| **LGDN (ours)** | | 5.2M | **34.3** | **62.5** | **72.2** | **3.0** | **34.7** | **60.8** | **70.4** | **3.0** |
| **LGDN$^\dagger$ (ours)** | | 15.2M | 39.8 | 65.2 | 77.0 | 2.0 | 39.2 | 66.4 | 76.1 | 3.0 |
| **1k-A Split:** | | | | | | | | | | |
| MMT [13] | ✓ | >100M | 26.6 | 57.1 | 69.6 | 4.0 | 27.0 | 57.5 | 69.7 | 3.7 |
| Support Set [40] | ✓ | >100M | 30.1 | 58.5 | 69.3 | 3.0 | 28.5 | 58.6 | 71.6 | 3.0 |
| TACo [52] | ✓ | >100M | 28.4 | 57.8 | 71.2 | 4.0 | | | | |
| Frozen in Time [2] | | 5.5M | 31.0 | 59.5 | 70.5 | 3.0 | - | - | - | - |
| **LGDN (ours)** | | 5.2M | **38.9** | **65.7** | **76.5** | **2.0** | **37.9** | **65.4** | **76.0** | **2.0** |
| **LGDN$^\dagger$ (ours)** | | 15.2M | 43.7 | 71.4 | 80.4 | 2.0 | 42.6 | 71.6 | 80.6 | 2.0 |

Table 4: Results on the VATEX test set.

| Method | R@1 | R@5 | R@10 | MdR |
|---|---|---|---|---|
| VSE [22] | 28.0 | 64.3 | 76.9 | 3.0 |
| VSE++ [10] | 33.7 | 70.1 | 81.0 | 2.0 |
| Dual [37] | 31.1 | 67.4 | 78.9 | 3.0 |
| HGR [5] | 35.1 | 73.5 | 83.5 | 2.0 |
| Support Set [40] | 45.9 | 82.4 | 90.4 | 1.0 |
| **LGDN (ours)** | **57.1** | **87.5** | **93.6** | **1.0** |
| **LGDN$^\dagger$ (ours)** | 61.0 | 90.2 | 95.1 | 1.0 |

Table 5: Results on the MSVD test set.

| Method | R@1 | R@5 | R@10 | MdR |
|---|---|---|---|---|
| VSE++ [10] | 15.4 | 39.6 | 53.0 | 9.0 |
| Multi. Cues [37] | 20.3 | 47.8 | 61.1 | 6.0 |
| CE [30] | 19.8 | 49.0 | 63.8 | 6.0 |
| Support Set [40] | 28.4 | 60.0 | 72.9 | 4.0 |
| Frozen in Time [2] | 33.7 | 64.7 | 76.3 | 3.0 |
| **LGDN (ours)** | **39.7** | **70.2** | **79.8** | **2.0** |
| **LGDN$^\dagger$ (ours)** | 43.2 | 73.3 | 82.4 | 2.0 |

Table 6: Results on the DiDeMo test set. $^*$ denotes using temporal labels of captions.

| Method | R@1 | R@5 | R@10 | MdR |
|---|---|---|---|---|
| S2VT [44] | 11.9 | 33.6 | - | 13.0 |
| FSE [53] | 13.9 | 36.0 | - | 11.0 |
| CE [30] | 16.1 | 41.1 | - | 8.3 |
| ClipBERT [25]$^*$ | 20.4 | 48.0 | 60.8 | 6.0 |
| Frozen in time [2]$^*$ | 34.6 | 65.0 | 74.7 | **2.0** |
| **LGDN (ours)** | **44.1** | **71.9** | **82.3** | **2.0** |
| **LGDN$^\dagger$ (ours)** | 47.8 | 76.2 | 83.3 | 2.0 |

Table 7: Results on MSRVTT-QA. $^*$ denotes utilizing large-scale VideoQA datasets.

| Method | #PT Pairs | Acc |
|---|---|---|
| Heterogeneous Memory [11] | - | 33.0 |
| HCRN [24] | - | 35.6 |
| SSML [1] | 100M | 35.1 |
| ClipBERT [25] | 5.6M | 37.4 |
| Just Ask$^*$ [51] | 69.0M | 41.5 |
| **LGDN (ours)** | 5.2M | **42.4** |
| **LGDN$^\dagger$ (ours)** | 15.2M | **43.1** |

## 4.3 Comparison to the State-of-the-Arts

We first report the text-video retrieval results on MSR-VTT with three data partitions in Table 3. It can be observed that: (1) our LGDN outperforms all previous works by large margins. Particularly, as compared with the most recent model Frozen in Time [2], our LGDN achieves an improvement of 7.9% (38.9% vs. 31.0%) for Text-to-Video R@1 on the MSR-VTT 1k-A test set. (2) Our LGDN also outperforms methods utilizing extra modalities (e.g., motion and audio) or those pre-trained on extremely-large video data (e.g., HowTo100M). (3) When leveraging a much larger pre-training (image-text) dataset, our LGDN (marked with $^\dagger$) achieves significant improvements.

To demonstrate the robustness of our model, we also evaluate it on VATEX, MSVD, and Didemo in Tables 4–6, respectively. Due to limited space, only text-to-video retrieval is considered here. For VATEX (Table 4), our LGDN significantly outperforms the state-of-the-art method Support Set which is trained on an order of magnitude more data. Our LGDN still performs the best on MSVD (Table 5) and Didemo (Table 6). Particularly, in the Didemo dataset, each description is

Table 8: Further evaluation results by directly applying our SFP mechanism to Clip4CLIP [34] (re-implemented) for video-text retrieval on the MSR-VTT 1k-A test set. * denotes the ensemble results of Clip4CLIP and Clip4CLIP+SFP.

| Method | Text-to-Video Retrieval | | | | | Video-to-Text Retrieval | | | | |
|---|---|---|---|---|---|---|---|---|---|---|
| | R@1 | R@5 | R@10 | MdR | MnR | R@1 | R@5 | R@10 | MdR | MnR |
| Clip4CLIP [34] | 44.9 | 71.8 | 81.7 | 2.0 | 14.2 | 46.2 | 73.9 | 84.3 | 2.0 | 10.8 |
| + SFP | 45.3 | 73.0 | 83.4 | 2.0 | 13.4 | 47.6 | 75.5 | 85.3 | 2.0 | 9.6 |
| + SFP* | **47.2** | **73.4** | **83.9** | **2.0** | **13.0** | **48.1** | **76.7** | **86.1** | **2.0** | **9.3** |

Table 9: Results by applying SFP to different sampling techniques on the MSR-VTT 1kA test set.

| Sampling | w\o SFP (R@SUM) | | | w\SFP (R@SUM) | | |
|---|---|---|---|---|---|---|
| | 4 frames | 8 frames | 16 frames | 4 frames | 8 frames | 16 frames |
| Random Sampling | 164.7 | 169.5 | 172.8 | 168.0 | 174.3 | 179.4 |
| Dense Uniform | 166.5 | 171.3 | 174.3 | 173.1 | 179.4 | 180.6 |
| Sparse Sampling | 168.0 | 171.9 | 173.7 | 179.1 | 180.3 | 181.1 |

Table 10: Ablation study for relevance score estimator. Retrieval results are reported on the MSR-VTT 1k-A test set. Random / SFP: $N_{salient}$ frames are sampled randomly/by our SFP mechanism. # Frames: $N_{salient}$ / N for local / global alignment.

| Method | Strategy | # Frames | Text-to-Video Retrieval | | | | Video-to-Text Retrieval | | | | R@SUM |
|---|---|---|---|---|---|---|---|---|---|---|---|
| | | | R@1 | R@5 | R@10 | MdR | R@1 | R@5 | R@10 | MdR | |
| All | Random | 16/16 | 35.7 | 63.8 | 74.2 | 3.0 | 35.4 | 63.8 | 74.9 | 3.0 | 347.8 |
| Random | Random | 2/16 | 34.1 | 62.1 | 73.4 | 3.0 | 34.3 | 61.6 | 74.0 | 3.0 | 339.5 |
| SimDot | SFP | 2/16 | 37.4 | 65.0 | 76.4 | 3.0 | 37.2 | 65.1 | 75.4 | 2.0 | 356.5 |
| Momentum | SFP | 2/16 | 38.1 | **65.8** | 76.4 | 2.0 | 37.9 | 65.4 | 75.9 | 2.0 | 359.5 |
| CrossMom | SFP | 2/16 | 38.4 | 65.4 | 76.5 | 2.0 | 37.9 | 65.3 | **76.2** | 2.0 | 359.7 |
| Collaborative | SFP | 2/16 | **38.9** | 65.7 | **76.5** | **2.0** | 37.9 | **65.4** | 76.0 | **2.0** | **360.4** |

annotated with localization information, in other words, annotations may only be aligned with the localized moments, thus causing the noise problem as many methods utilize all frames as the input. Recent works exploit temporal labels of captions to alleviate the noise problem and achieve higher performance. However, even without considering this, our LGDN still largely outperforms the most recent method Frozen [2], further demonstrating the effectiveness of our LGDN.

To show the general applicability of our LGDN, we evaluate our LGDN on the VideoQA task in Table 7. Even without utilizing large-scale video datasets devoted to the VideoQA task, our LGDN outperforms all competitors, validating the effectiveness of our LGDN in VideoQA. In addition, to reveal the critical importance of solving the noise issue for video-language modeling, we directly apply the SFP mechanism to the latest model CLIP4Clip [34] in Table 8. We find that applying the SFP mechanism brings boost to Clip4CLIP. The ensemble mechanism further improves the results, indicating that the proposed SFP mechanism is complementary to the baseline.

## 4.4 Additional Results

**Applying SFP to Different Frame Sampling Techniques.** Note that our SFP mechanism must be combined with a frame sampling technique since we adopt a two-stage sampling strategy in this paper. Thus, we apply our SFP mechanism to three frame sampling techniques: Sparse Sampling, Random Sampling, and Dense Uniform (equally interval sampling). The obtained results on the MSR-VTT 1kA test set are provided in Table 9. It can be observed that our SFP significantly boosts different sampling strategies, further demonstrating the general applicability of our SFP mechanism.

**Expansion of Relevance Score Estimator.** In Sec. 3.3, we have proposed four relevance score estimators for the LSFM module. To find out which is the best, we present the ablation study results for different relevance score estimators in Table 10. We can see a large gap between SFP and random sampling (w/o SFP), directly demonstrating the effectiveness of the proposed SFP mechanism. Meanwhile, both Momentum and CrossMom outperform SimDot, suggesting that introducing momentum encoder is beneficial to relevance score estimation. Collaborative that combines Momentum and CrossMom generally leads to further improvements.

Table 11: The model capacity of different recent methods on the MSR-VTT 1kA test set.

| Methods | Visual Encoder | Lingual Encoder | Fusion Layer | Total | R@SUM |
|---|---|---|---|---|---|
| TACo [52] | 155M | 110M | 14M | 279M | 157.4 |
| Support Set [40] | 136M | 220M | - | 356M | 157.9 |
| Frozen in Time [2] | 114M | 66M | - | 180M | 161.0 |
| LGDN (global) | 93M | 55M | - | **148M** | 164.6 |
| LGDN (ours) | 93M | 55M | 68M | 215M | **181.1** |

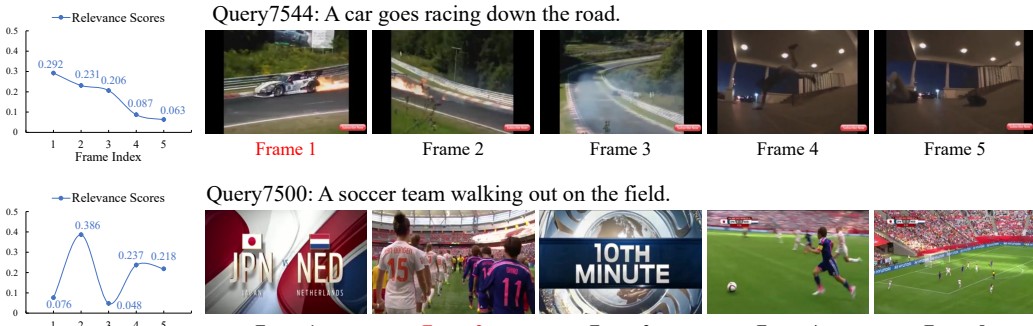

Figure 4: Visualization of our LGDN on the MSR-VTT test set. We uniformly sample 5 frames from each video and the red denotes salient frames selected by our SFP mechanism. We find that our SFP mechanism indeed filters out the unmatched/redundant frames under the language supervision.

**Model Capacity.** We also provide the detailed comparison to other methods in terms of model capacity and R@SUM (on the MSR-VTT 1kA test set) in Table 11. It can be clearly seen that: (i) When fusion layers are not used (i.e., only global alignment is adopted), our LGDN (global) outperforms the state-of-the-art method Frozen in Time [2], but with much less model parameters. (ii) Our full LGDN performs much better than all the competitors, but its parameter number (215M) is still comparable to that of Frozen in Time (180M) and even significantly smaller than those of the other competitors. These observations suggest that the performance gains obtained by our LGDN is not due to utilizing more model parameters.

### 4.5 Visualization Results

We provide visualization of our LGDN in Figure 4. We uniformly sample 5 frames from each video and provide relevance scores of 5 frames on the left, and the red ones denote salient frames selected by the SFP mechanism. It can be seen that: (1) Although the holistic video is semantically related to the paired text, there still exist noisy frames (e.g., the transition in Frame 1 and Frame 3 of Query7500) and unrelated frames (e.g., in Frame 4 and Frame 5 of Query7544, a man is rolling while the paired text is 'a car goes racing down the road'). (2) The relevance scores obtained from the SFP mechanism correctly measure the consistency between each frame and the paired text, which indeed helps our LGDN to precisely filter out noisy information for better video-language modeling.

## 5 Conclusion

In this work, we propose a novel Language-Guided Denoising Network (LGDN) for video-language modeling, which can dynamically filter out the unmatched or redundant frames under the language supervision and thus maintain only 2–4 salient frames per video for cross-modal token-level alignment. Extensive experiments on five public datasets show that our LGDN outperforms the state-of-the-arts by large margins. In the future, we will consider aggregating temporal information on salient frames and apply our approach to more challenging video-language tasks (e.g., video grounding).

## Acknowledgments and Disclosure of Funding

This work was supported in part by National Natural Science Foundation of China (61976220 and 61832017), Beijing Outstanding Young Scientist Program (BJJWZYJH012019100020098), and the Research Seed Funds of School of Interdisciplinary Studies, Renmin University of China.

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
