# LGDN: Language-Guided Denoising Network
# for Video-Language Modeling
# – Supplementary Material –

**Haoyu Lu**[1,2]   **Mingyu Ding**[3]   **Nanyi Fei**[1,2]   **Yuqi Huo**[4]   **Zhiwu Lu**[1,2,*]

[1]Gaoling School of Artificial Intelligence, Renmin University of China, Beijing, China
[2]Beijing Key Laboratory of Big Data Management and Analysis Methods
[3]The University of Hong Kong, Pokfulam, Hong Kong
[4]JD Corporation, Beijing, China
{lhy1998, luzhiwu}@ruc.edu.cn

## 1   Limitations and Potential Negative Societal Impacts

**Limitations.**   The key idea of our LGDN is to propose the SFP mechanism to filter out noisy/redundant frames for fine-grained semantic alignment, along with MVCL for capturing global temporal information. In most downstream tasks, these two modules are complementary to each other. And we also observe that only a few salient frames (e.g., 2 ones) are enough for most downstream tasks, and thus we do not consider aggregating temporal information across salient frames. However, the SFP mechanism may need to be slightly changed when facing specified scenarios (e.g., long-term complicated videos over 30 minutes that highly rely on temporal information). On the one hand, we could adjust the weights between the two modules (MVCL and SFP) according to the situation. On the other hand, we can split the full video into several clips (e.g., 3 minutes per clip), apply our SFP mechanism on each clip, and obtain the salient frames from all clips. In this way, we could consider aggregating temporal information across salient frames.

**Potential Negative Societal Impacts.**   Video-language learning, especially large-scale video-language modeling, has developed rapidly over the past few years and led to the greatest advance in search engines, video recommendation, and multimedia data management. Despite its effectiveness, existing video-language pre-training models still face possible risks. As these models often rely on a large amount of web data, they may acquire biases or prejudices (especially in search engines and recommendation systems), which must be properly addressed before model deploying.

## 2   More Implementation Details

**Two-Stage Sampling Strategy.**   The sampling strategy of our LGDN has two stages as shown in Figure 1: (1) We first adopt sparse sampling to sample 16 frames from each video before feeding them into the LGDN, which is the same as ClipBERT. (2) We further utilize salient sampling (SFP) to select a few salient frames (from 16 frames per video) before fusion layers.

**Details of Network Architecture.**   We adopt ViT-B/16 [3] as our frame encoder and the first 6 layers of BERT-base [2] as our text encoder. The dimensions of the output vectors of the frame and text tokens are both $N_{seq} \times 768$, where $N_{seq}$ is the sequence length. For each frame/text, the final output vector of the [CLS] token is used as the frame/text embedding. We utilize one Transformer layer (with 768 hidden units and 12 heads) as the temporal module to aggregate the frame embeddings to obtain the video embedding. We then utilize a single fully-connected layer for each modality to project the frame/video/text embeddings to the joint cross-modal space. The final dimensions of the frame and text embeddings are 256. We further apply a 6-layer cross-attention Transformer with an additional cross-attention module in each layer as our multi-modal encoder as in LXMERT [8]

---

[*]The corresponding author.

36th Conference on Neural Information Processing Systems (NeurIPS 2022).

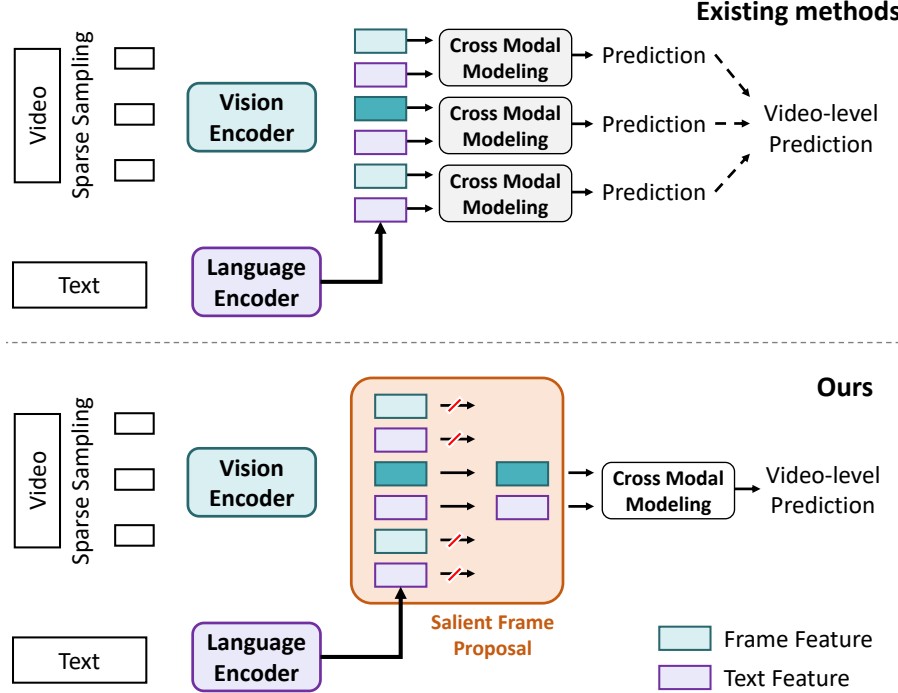

Figure 1: Comparison between the sparse sampling paradigm (e.g., ClipBERT, top) and our proposed LGDN (bottom). Different from most existing methods that utilize all extracted video features, LGDN dynamically filters out the unmatched or redundant frames under the language supervision.

and ALBEF [6], where the network parameters are initialized from the last 6 layers of BERT-base. Each layer of the cross-attention Transformer consists of a self-attention sub-layer, a cross-attention sub-layer, and a feed-forward sub-layer with 768 hidden units and 12 heads.

**Downstream Datasets.** (1) **MSR-VTT** [11] is a popular video-text dataset with three data partitions. The full split is the official partition which uses 6,513/497/2,990 videos for training/validation/testing. The 1k-A split is a widely-used partition with 9,000/1,000 videos for training/testing. Recent works also apply the 7k-1k split, which uses 7,000/1,000 videos for training/testing. All three data partitions are considered in our experiments. (2) **MSVD** [11] contains 80K descriptions for 1,970 videos from YouTube. Following Frozen in Time [1], we employ the standard split with 1,200 videos for training and 670 videos for testing. (3) **DiDeMo** [4] consists of 10K videos and 40K sentences. Each sentence includes the temporal localization information. Following Frozen in Time [1], we conduct the paragraph-to-video retrieval task, where all descriptions in the same video are concatenated into a single description. (4) **VATEX** [9] is composed of 34,911 videos. We use 25,991/1,500/1,500 videos for training/validation/testing, following the split in Support Set [7]. (5) **MSRVTT-QA** [10] is a widely-used video-question answering dataset. We employ the standard split as in ClipBERT [5].

**Resources Used.** It takes around 3 / 9 days to pre-train LGDN (5.2M / 15.2M) with 16 Tesla V100 GPUs. For each downstream task, it takes about 5-15 hours with 8 Tesla V100 GPUs.

## 3 More Experimental Results

**Effect of SFP mechanism.** To further demonstrating the effectiveness of SFP mechanism, we conduct experiments considering model variants that select $N_{salient}$ salient frames from $N = 16$ frames by our SFP or just randomly select $N_{salient}$ frames (denoted as w/o SFP) for token-level alignment in Figure 2. We sample $N = 16$ frames from each video and evaluate different model variants that use $N_{salient} \in \{1, 2, 3, 4, 8, 16\}$ frames for token-level alignment. Note that when $N_{salient} = 16$, sampling by our SFP degrades to w/o SFP. As expected, when randomly selecting frames like most existing methods, adding more frames does bring better results. However, utilizing only $N_{salient} = \{2, 3, 4\}$ salient frames filtered by our SFP significantly outperforms random

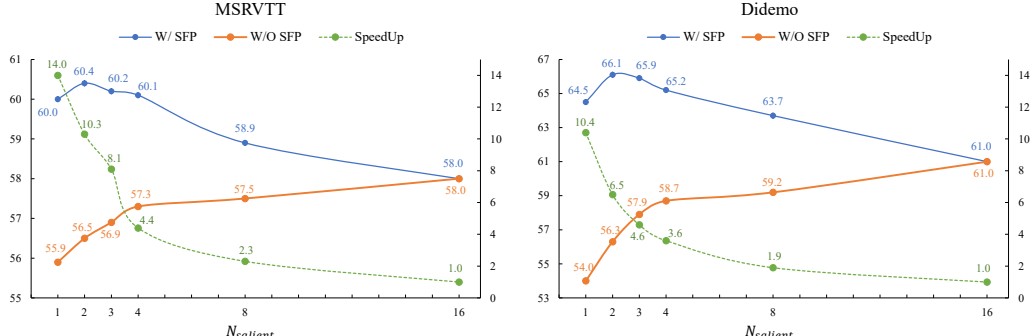

Figure 2: Effect of value change of $N_{salient}$ on the performance of our LGDN for text-to-video retrieval on the MSR-VTT / Didemo test set. w/ SFP: frames are filtered by our SFP mechanism. w/o SFP: frames are randomly selected. SpeedUp: the speedup over $N_{salient} = 16$.

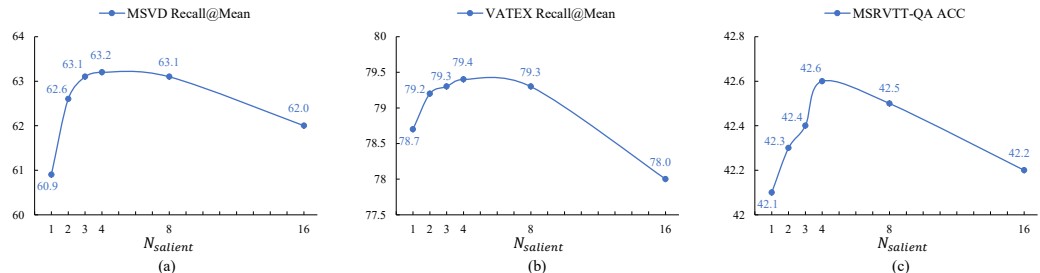

Figure 3: (a) Effect of value change of $N_{salient}$ on the performance of our LGDN for text-to-video retrieval on the MSVD test set. (b) Effect of value change of $N_{salient}$ on the performance of our LGDN for text-to-video retrieval on the VATEX test set. (c) Effect of value change of $N_{salient}$ on the performance of our LGDN for video question answering on the MSRVTT-QA val set.

Table 1: Effect of value change of memory bank size on the performance of our LGDN model. Video-text retrieval results are reported on the MSR-VTT 1k-A test set.

| Bank Size | Text-to-Video Retrieval | | | Video-to-Text Retrieval | | |
|---|---|---|---|---|---|---|
| | R@1 | R@5 | MdR | R@1 | R@5 | MdR |
| 1,200 | 36.0 | 65.2 | 3.0 | 36.2 | 64.8 | 3.0 |
| 2,400 | 36.9 | 64.6 | 3.0 | 37.0 | 64.8 | **2.0** |
| 4,800 | 37.7 | **65.7** | 3.0 | 37.8 | 65.0 | **2.0** |
| 9,600 | **38.9** | **65.7** | **2.0** | 37.9 | **65.4** | **2.0** |
| 19,200 | 38.3 | 64.9 | 3.0 | 37.5 | 64.3 | 3.0 |

sampling (i.e., w/o SFP), and even outperforms utilizing all 16 extracted frames. This suggests that our SFP mechanism not only selects correct salient frames but also alleviates the noise problem.

We also present the experiment results on other public datasets in Figure 3. Though the best parameters $N_{salient}$ are not quite the same on different datasets ($N_{salient} = 2$ for MSRVTT, Dedimo; $N_{salient} = 4$ for MSVD, VATEX, MSRVTT-QA), it can be observed that SFP mechanism significantly improves the baseline. Meanwhile, the performance changes among the three parameters ($N_{salient} \in \{2, 3, 4\}$) are very marginal, which further verifies the robustness and effectiveness of our LGDN.

**Effect of Value Change of Memory Bank Size.** In Table 1, we show the influence of different values of the memory bank size $N_m$ on the performance of our LGDN. With the increase of memory bank size $N_m$, the performance of our LGDN begins with an increase, indicating that the introduction of large-scale negatives for contrastive learning indeed brings performance improvements. However, when $N_m$ becomes too large ($> 9,600$), the performance drops slightly. One possible reason is that too large memory bank size may introduce more hard negative samples, which makes it harder to learn a good vision-language representation. Our LGDN thus performs the best at $N_m = 9,600$.

Table 2: The speed and memory cost of different sampling strategies on the Deidemo test set.

| Sampling Strategy | Speedup | Memory Cost | R@SUM |
|---|---|---|---|
| Sparse sampling | 1.0x | 1.0x | 183.0 |
| Salient sampling ($N_{salient} = 1$) | 10.4x | 0.60x | 193.5 |
| Salient sampling ($N_{salient} = 2$) | 6.5x | 0.62x | 198.3 |
| Salient sampling ($N_{salient} = 4$) | 3.6x | 0.68x | 195.6 |

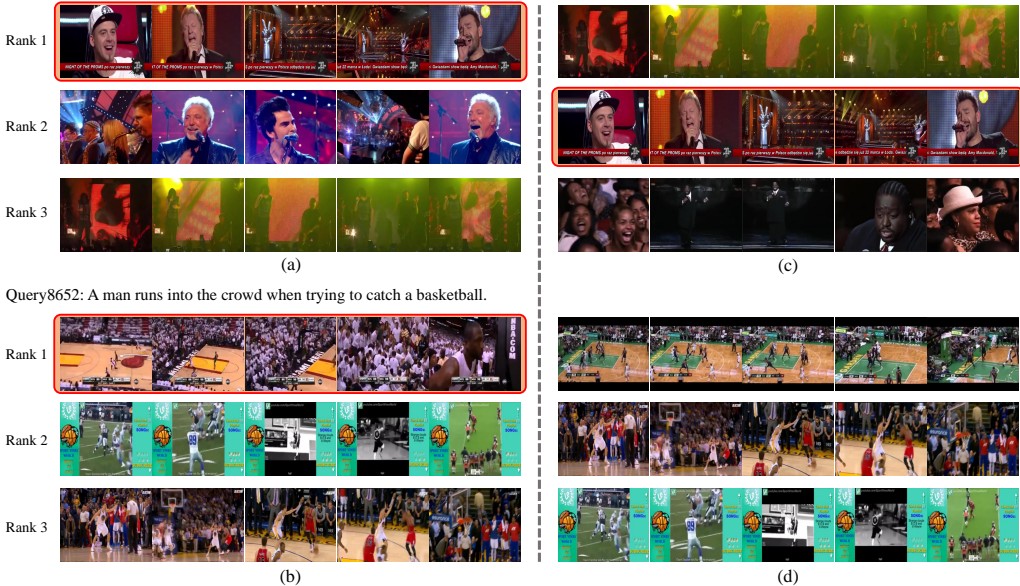

Figure 4: Top-3 text-to-video retrieval results on the MSR-VTT 1k-A test set (correct result is highlighted in red). The left are videos ranked by our LGDN, while the right are results from our model without using the SFP mechanism.

**Speed and Memory Cost.** We present the speed and memory cost on the Didemo test set in Table 2. For fair comparison, all experiments are conducted on 8 Tesla-V100 GPUs with mini-batch size 24. It can be seen that our salient sampling strategy is obviously faster and costs less memory, as compared with sparse sampling (utilizing all 16 frames for feature extraction and multi-modal fusion).

## 4 Visualization Results

We provide visualization examples in Figure 4. Figures 4(a)-(b) are the retrieved results by our LGDN and Figures 4(c)-(d) are the retrieved results by our model without using the SFP mechanism. We can see from Figures 4(a)-(b) that although the target videos have noisy frames (e.g., in the first frame of rank 1 video of Figure 4(a), the man is laughing; the last frame of rank 1 video of Figure 4(b) is the close up of the player who run into the crowd), LGDN precisely retrieves the ground-truth. In Figure 4(c), LGDN without using SFP also retrieves the corresponding video where the people are singing to the audience, however, in rank 1 video, there are a woman and a man both singing, which is incorrect to the query "A man". In Figure 4(d), LGDN without using SFP only retrieves video corresponding to basketball and crowd. These examples indicate that the noisy information misleads cross-modal modeling and our LGDN with SFP can help to alleviate it.

Further, we provide more visualization results obtained by our LGDN in Figures 5-6. We uniformly sample 6 frames from each video, among which the red ones denote salient frames selected by the SFP mechanism. It can be clearly observed that: (1) Although the holistic video is semantically related to the paired text, there still exist noisy frames (e.g., the transition in Frame 3-6 of Query7466) and unrelated frames (e.g., Frame 4-6 of Query7468, Frame 1-5 of Query7586, Frame 2-3 of Query8069, and Frame 2-5 of Query8265). (2) The salient frames obtained from the SFP mechanism correctly represent the semantic information of the video given the paired text, which indeed helps our LGDN to precisely filter out noisy information for better video-language modeling.

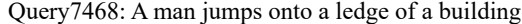

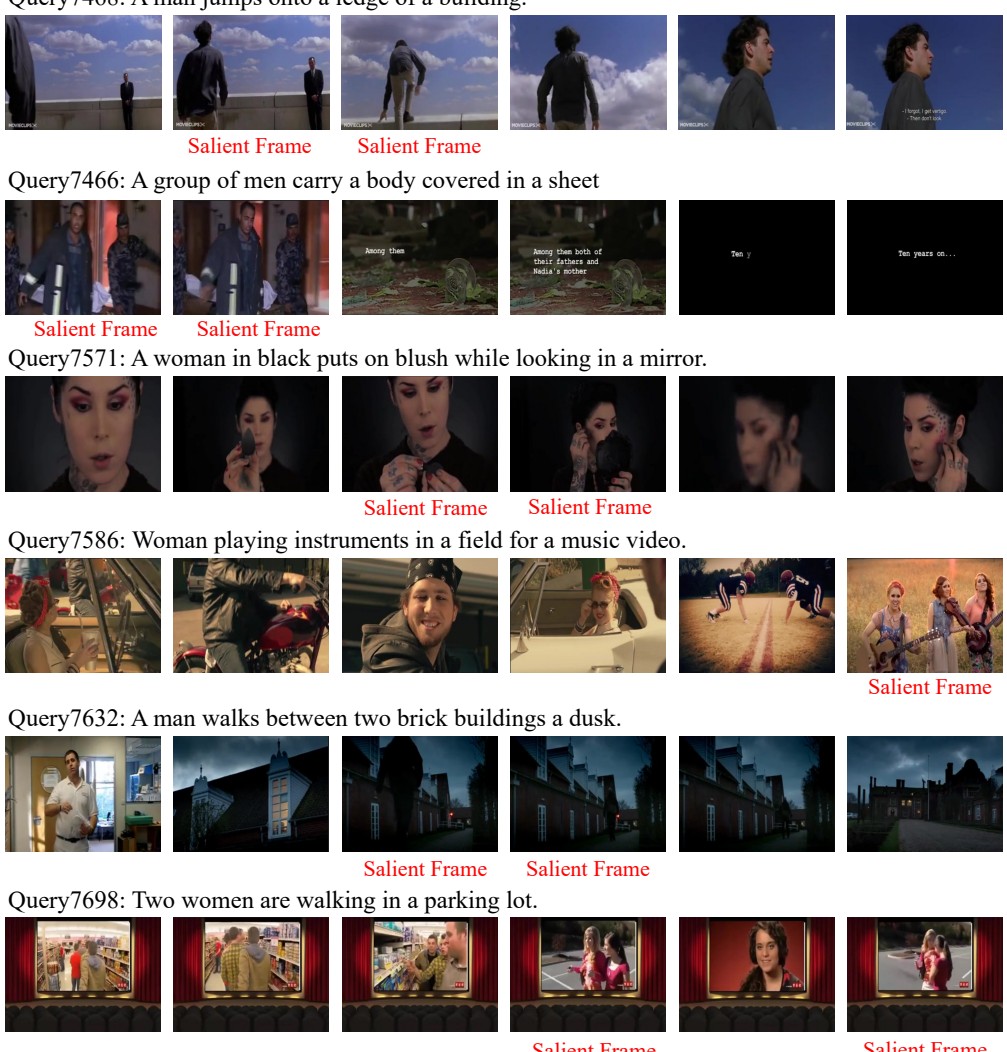

Figure 5: Visualization of salient frames obtained by our LGDN on the MSR-VTT test set. We uniformly sample 6 frames from each video, among which the red ones denote the salient frames selected by our SFP mechanism. We find that our SFP mechanism indeed filters out the unmatched/redundant frames under the language supervision.