# OpenReview forum: "LGDN: Language-Guided Denoising Network for Video-Language Modeling"
_NeurIPS.cc/2022/Conference — NeurIPS 2022 Accept_

### Official Review · Reviewer_UdJY · 2022-07-09

**Rating:** 5
**Confidence:** 4
**Soundness:** 3 good
**Presentation:** 3 good
**Contribution:** 2 fair

**Summary:**

The paper proposes a Language-Guided Denoising Network (LGDN) for video-language modeling, which can deal with the noisy information in video frames. LGDN dynamically filters out the misaligned or redundant frames under the language supervision and obtains only 2–4 salient frames per video for cross-modal token-level alignment. Extensive experiments on five public datasets show that LGDN outperforms the state-of-the-arts by large margins.

**Questions:**

1. Table 2 shows the improvement of $\mathcal L_{MVCL}$ is marginal. Is it necessary to the LGDN model? Can MIL-NCE be used to compute the $\mathcal L_{MVCL}$?
2. The authors claim that the proposed LGDN outperforms the state-of-the-arts by large margins. However, as shown in Table 8, LGDN performs worse than Clip4CLIP. This point needs further discussion.
3. Since the pre-training with CC12M brings significant improvements, it would be nice to compare some SOTA methods under the same pre-training settings.


**Limitations:**

The authors did not discuss the limitations and potential negative societal impact of their work.

**Strengths And Weaknesses:**

Strengths:
The paper is well organized and easy to read. The noise issue in video-text retrieval task studied by this work is practical, and the proposed salient frame proposal mechanism seems simple and effective. Extensive experiments are performed on video-text datasets and the experimental results are promising.

Weaknesses:
The paper needs more analysis and ablation studies to support its network architecture and proposals. For example, in-depth comparison between salient frame proposal (SFP) mechanism and sparse sampling or other frame sampling techniques. Also, some experimental results (e.g., Effect of SFP mechanism) in supplementary material can be included in the paper.

---

> ### Author Response · Authors · 2022-08-01
> **Response to Reviewer UdJY**
>
>
> Thank you for the positive comments and insightful suggestions.
>
> **Q1. [Provide more comparison between salient frame proposal (SFP) mechanism and sparse sampling or other frame sampling techniques.]** \
> **A:** Thanks for the suggestion. Note that our SFP mechanism must be combined with a frame sampling technique since we adopt a two-stage sampling strategy in this paper (see our **response to Q1 of Reviewer HH2K**). We have already presented the comparison between sparse sampling (used in ClipBERT) and sparse sampling + SFP in Figure 3 of the main paper. Here, we apply our SFP mechanism to another two frame sampling techniques: Random sampling and Dense Uniform (equally interval sampling). The obtained results on the MSR-VTT 1kA test set are provided in the following table. It can be observed that our SFP significantly boosts different frame sampling strategies, further demonstrating the general applicability of our SFP mechanism. In addition, we have added these results in **Table 3 of the supplementary material**. We also provide speedup and memory efficiency results compared with sparse sampling strategy in our **Response to Reviewer HH2K Q1**.
>
> |                 | R@SUM |   R@SUM    |    R@SUM   |   R@SUM     |    R@SUM   |    R@SUM   |
> |-----------------|:---:|:---:|:---:|:---:|:---:|:---:|
> |     #Frames     |       4       |   8   |   16  |    4   |   8   |   16  |
> | + SFP     |    no    |    no   |  no    | yes |    yes   |   yes    |
> | Random sampling |     164.7     | 169.5 | 172.8 |  168.0 | 174.3 | 179.4 |
> |  Dense Uniform  |     166.5     | 171.3 | 174.3 |  173.1 | 179.4 | 180.6 |
> | Sparse sampling |     168.0     | 171.9 | 174.0 |  179.1 | 180.3 | 181.1 |
>
> **Q2. [Some experimental results (e.g., Effect of SFP mechanism) in supplementary material can be included in the paper.]** \
> **A:** Thanks for pointing this out. Due to the limited space, we have to put them in the supplementary material. We are glad to place them  (including the above comparison results) in the main paper in any other form of publication (e.g., Arxiv).
>
> **Q3. [Table 2 shows the improvement of MVCL is marginal. Is it necessary for the LGDN model? Can MIL-NCE be used to compute the MVCL?]** \
> **A:** Good question!
> 1. MVCL is mainly designed for capturing global temporal information, and thus it only leads to marginal improvement for local alignment (**335.2 vs. 336.1** in Table 2 of the main paper). However, in our downstream tasks, MVCL is found to be complementary to the local alignment and helps our LGDN achieve much higher performance (**352.9 vs. 360.4** in Table 2 of the main paper).
> 2. In our downstream datasets, each video annotated carefully by human is holistically (semantically) consistent with the paired caption, and thus a traditional NCE loss is suitable. However, if we pre-train our LGDN on web datasets with noisy narration (e.g., HowTo100M), a MIL-NCE loss is thus needed.
>
> **Q4. [Why LGDN (in Table 3) performs slightly worse than Clip4CLIP (in Table 8).]** \
> **A:** Sorry for the confusion. Our LGDN is pre-trained on 5.2M/15.2M image-text pairs, while Clip4CLIP applies OpenAI's CLIP [2] as the backbone pre-trained on 400M image-text pairs (**75.9x/25.3x larger** than ours). Nevertheless, even with much less pre-training data, our LGDN (in Table 3 of the main paper) still performs comparably w.r.t. Clip4CLIP (in Table 8 of the main paper), indicating the effectiveness of our LGDN. Moreover, as shown in Table 8 of the main paper, when we apply CLIP as our backbone, our LGDN significantly outperforms Clip4CLIP, which further demonstrates the general applicability of our LGDN.
>
> **Q5. [It would be nice to compare some SOTA methods with CC12M.]** \
> **A:** Good suggestion. To the best of our knowledge, we have for the first time applied CC12M to video-language pre-training (and also achieved great success) in this paper. Due to the limited computing resources and rebuttal time, we could not reproduce other competitors with CC12M as the pre-training data. We will do it in our future works.
>
> **Q6. [Lack of limitations and potential negative societal impacts.]** \
> **A:** Thanks. We have added them in Section 1 of the supplementary material.
>
> [2] Alec Radford, et al. "Learning Transferable Visual Models From Natural Language Supervision." ICML 2021.
>
>
> Thanks for your time and effort again! For any other questions, please feel free to let us know during the rebuttal window.

---

> ### Author Response · Authors · 2022-08-08
> **Looking forward to your post-rebuttal feedback**
>
> Dear Reviewer UdJY,
>
> Thanks again for your constructive suggestions, which have helped us improve the quality and clarity of the paper!
>
> In our previous response, we have carefully studied your comments and made detailed responses. As the discussion period will end soon in 2 days, we are happy to provide any additional clarifications that you may need. Please do not hesitate to contact us if there are other clarifications we can offer. We appreciate your suggestions.
>
> Thanks for your time and efforts!
>
> Best, \
> Authors

---

### Official Review · Reviewer_HH2K · 2022-07-09

**Rating:** 6
**Confidence:** 4
**Soundness:** 3 good
**Presentation:** 3 good
**Contribution:** 3 good

**Summary:**

This paper introduces LGDN, which utilizes a self-taught mechanism to select salient frames to train the fusion layers. The vision and language encoder still use all frames and texts to train, and their goal is to learn representations while computing relevance scores for the frame selection. The whole mechanism by contrastive learning helps to filter out noisy information in the last few layers, boosting performance while reducing computation costs. They also demonstrate better empirical results over the baselines.

**Questions:**

1. ClipBERT samples frames before feeding them into the model while your approach does the sampling only before fusion layers, so I wonder about the memory efficiency and speed of LGDN compared to ClipBERT, Frozen in Time, and other methods.
2. I understand that the SFP might address the misalignment issue, but how it can help to ease the redundancy in the video. If you compute the relevance score between text and frames, it's likely that similar frames have similar scores, and I am not sure what can prevent LGDN to select the other frames that are similar to the salient frame. I feel like some diversity loss might be necessary to address this problem.
3. What's the model size of other baselines? LGDN is composed of vision, language, and fusion Transformers while Frozen in Time doesn't have a fusion Transformer, making me concerned if the performance gain somewhat comes from the bigger models. For better comparison, you can try to use lesser layers in vision and language encoders.
4. What are global and local alignments used in the inference phase? I didn't find an explanation in the paper. Why does using global inference generates such low performance?
5. The salience sampling breaks the temporal dependency of the video (compared to uniform sampling), which is an important characteristic to solve most video tasks. Do you think this sampling will prevent LGDN from applying to other video tasks that highly depend on temporal information?

**Limitations:**

The paper's writing can be improved. The method has many components, and I was confused about which part addresses what problem at the beginning. But in general, I still understand the idea and contribution of the paper. My main concern is about the fairness of the experiments and some questions about the technical approach (refer to Questions). I will raise the score if those problems are addressed.

**Strengths And Weaknesses:**

1. Pros:

* The paper proposes an effective method to select frames from noisy video, leading to an improvement in both fine-tuned accuracy and computation efficiency.

* To address the noisy problem in video is novel to me, it might become a new approach to constructing multi-modal models. There is no perfect and exact matching between any kinds of pair data so that the impact can be extended to other domains.

2. Cons:

* The empirical results are not fully convincing to me. I think it would be better if the authors can report the model capacity of each method (see Questions 3).

* The central claim of this paper is that noisy video might harm the training, but the model still highly relies on learning on noisy data as the vision and language encoder still use full video to learn how to match the relevance between language and frames.

* It seems to me that the redundancy issue has not been addressed by LGDN (see Question 2), and the sampling might introduce another problem (see Question 3).

---

> ### Author Response · Authors · 2022-08-01
> **Response to Reviewer HH2K - Part II**
>
> **Q3. [The model size compared with other methods. For better comparison, LGDN can use lesser layers in vision and language encoders.]** \
> **A:** Sorry for the confusion. Our full LGDN model consists of ViT-Base (with 12 Transformer layers as the visual encoder) and BERT-Base (with the first 6 Transformer layers as the lingual encoder and the last 6 as multi-modal fusion layers). We have clarified this network structure in Section 2 of the supplementary material. Moreover, we also provide detailed comparison to other methods in terms of model capacity and R@SUM (on the MSR-VTT 1kA test set) in the following table. We have added these results in **Table 2 of the supplementary material**.
> |   Methods     | Visual Encoder | Lingual Encoder | Fusion Layer | Total |  R@SUM |
> |     :-       |       :-:      |      :-:        |      :-:     |  :-:  |  :-:   |
> |      UniVL   |   110M    |     110M      |     46M    |    266M    |   133.9   |
> |     TACo      |       155M     |      110M       |      14M     |  279M |  157.4 |
> |   Support Set  |       136M     |      220M       |       -      |  356M |  157.9 |
> |    Frozen in Time     |       114M      |     66M       |       -      |  180M |  161.0 |
> | LGDN (global) |       93M      |      55M        |       -      |  **148M** |  164.6 |
> |  LGDN (ours)  |       93M      |      55M        |      68M     |  215M |  **181.1** |
>
> It can be observed that:
> - When fusion layers are not used (i.e., only global alignment is adopted), our LGDN (global) outperforms the state-of-the-art method Frozen in Time, but with fewer model parameters.
> - Our full LGDN performs much better than all the competitors, but its parameter number (215M) is still comparable to that of Frozen in Time (180M) and even significantly smaller than those of the other competitors. These observations suggest that the performance gains obtained by our LGDN are not due to utilizing more model parameters.
>
>
>
> **Q4. [What are global and local alignments used in the inference phase? Why does using global inference generates such low performance?]** \
> **A:**
> 1. Thanks for pointing this out. In the inference phase, the global similarity scores are first obtained by computing the dot product between text embeddings and video embeddings from MVCL (i.e., global alignment). Further, the token-level similarity scores are extracted from the LSFM module as described **in Section 3.5** (i.e., local alignment). We simply add these two scores (i.e., global+local alignment) for the final prediction.
> 2. The low performance of using global inference alone (see Table 2 of the main paper) can be explained as follows:
>     * to maintain temporal information (which is needed for video tasks), the global alignment directly takes all 16 sampled frames of each video as input, which does not consider the noise issue;
>     * the global alignment only adopts dot product as similarity scores before fusion layers, while the local alignment exploits token-level interaction for better performance through multi-modal fusion layers.
> However, as shown in Table 2 of the main paper, the global alignment still leads to significant improvements when it is combined with the local alignment (i.e., Ensemble).
>
> **Q5. [The salience sampling breaks the temporal dependency of the video (compared to uniform sampling), which is an important characteristic to solve most video tasks. Do you think this sampling will prevent LGDN from applying to other video tasks that highly depend on temporal information?]** \
> **A:** Good question! We agree that the salience sampling breaks the temporal dependency of the video. However, from Table 2 of the main paper, we can observe that even without considering the temporal information (video-level MVCL is not used), our LGDN still largely outperforms the state-of-the-arts (in Table 3 of the main paper). This indicates that the semantic correlation is as important as temporal information for most video tasks. Therefore, it is essential to exploit both semantic correlation and temporal information for video-language modeling. Inspired by this, we thus propose SFP for fine-grained semantic alignment and adopt MVCL in our LGDN for capturing global temporal information. In our downstream tasks, these two modules are found to be complementary to each other, and their fusion leads to better performance (see Table 2 of the main paper). This suggests that our salience sampling does not prevent LGDN from applying to other video tasks that highly depend on temporal information. More importantly, we can adjust the weights of these two modules to pay more attention to temporal information in this case.
>
>
> We wish that our response has addressed your concerns, and turns your assessment to the positive side. If you have any questions, please feel free to let us know during the rebuttal window. We appreciate your suggestions and comments! Thank you!

---

> ### Author Response · Authors · 2022-08-01
> **Response to Reviewer HH2K - Part I**
>
> Thank you for the constructive comments and suggestions.
>
> **Weaknesses:**
>
> **W1. [It would be better if the authors can report the model capacity of each method (see Questions 3).]** \
> **A:** Good suggestion. Please see our response to Q3 for detailed information.
>
> **W2. [LGDN still highly relies on learning on noisy data as the vision and language encoder still use full video to learn how to match the relevance between language and frames.]** \
> **A:** Sorry for the confusion. Our MVCL (utilizing full video) is mainly designed for capturing global temporal information. As for matching the relevance between language and frames, we introduce MFCL which utilizes the salient frames filtered by the SFP Mechanism in each video (instead of utilizing full video) to form a set of positive candidate pairs for frame-level MIL-contrastive learning. Please see Section 3.4 or Figure 2 in the main paper for more details.
>
> **W3. [The redundancy issue has not been addressed by LGDN (see Question 2), and the salient sampling might introduce another problem (see Question 5).]** \
> **A:** (1) The redundancy issue is discussed in our response to Q2. (2) Good question! And this is the reason why we propose MVCL for our LGDN in Section 3.2. Please see our response to Q5 for detailed information.
>
> **Questions:**
>
> **Q1. [ClipBERT samples frames before feeding them into the model while LGDN does the sampling only before fusion layers, which may cost more memory or be slower than other methods (e.g., ClipBERT, Frozen in Time).]** \
> **A:** Sorry for the confusion. The sampling strategy of our LGDN includes two stages:
> 1. We first adopt sparse sampling to sample 16 frames from each video before feeding them into the LGDN as introduced in Section 4.1 Implementation Details, which is the same as ClipBERT and Frozen in Time.
> 2. We further utilize salient sampling (SFP) to select a few salient frames (from 16 frames per video) before fusion layers.
>
> To clarify the two stages of our sampling strategy, we have added a schematic figure (i.e., **Figure 1**) in the supplementary material. Further, we present the speed and memory cost on the Didemo test set in the following table. For fair comparison, all experiments are conducted on 8 Tesla-V100 GPUs with mini-batch size 24. It can be seen that our salient sampling strategy is obviously faster and costs less memory, as compared with sparse sampling (utilizing all 16 frames for feature extraction and multi-modal fusion). We have added these results in **Table 1 of the supplementary material**.
>
>
> |                          |  Speedup  | Memory Cost |   R@SUM |
> |         :-        | :-:  |    :-: |:-:|
> |     Sparse sampling      |   1.0x    |     1.0x    |       183.0 |
> |  Salient sampling ($N_{salient}=1$) |   10.4x    |     0.60x   | 193.5 |
> |  Salient sampling ($N_{salient}=2$) |   6.5x    |     0.62x   | **198.3** |
> |  Salient sampling ($N_{salient}=4$) |   3.6x    |     0.68x   | 195.6 |
>
> In fact, we have already presented the speedup of our salient sampling strategy in Figure 3 (a-b) of the main paper, which is computed w.r.t. the slowest case $N_{salient} = 16$ (i.e., sparse sampling).
>
> **Q2. [How does SFP help to ease the redundancy in the video? Meanwhile, the other frames that are similar to the salient frame may also be selected by the SFP mechanism (some diversity loss might be necessary to address this problem).]** \
> **A:** Good questions!
> 1. Most recent works utilize sparse sampling to sample 16 frames per video for video-language modeling. As we have mentioned in our **response to Q1**, our SFP further selects a few salient frames (e.g., 2 ones) out of 16. The use of much fewer frames by two-stage frame sampling indeed contributes to easing the redundancy in the video.
> 2. In this work, the problem of sampling similar salient frames can **almost be ignored**. The sparse sampling strategy (16 frames per video) has alleviated the frame redundancy at the very beginning. Therefore, this problem is not necessary to be addressed in our current work. However, we agree that a well-designed diversity loss may help LGDN to handle more complex scenarios. Thanks for the valuable suggestion. We are working on it, but due to the limited time and the computing resources, we are expected to include the loss and experiments in our next revision.

---

> ### Author Response · Authors · 2022-08-07
> **Looking forward to your post-rebuttal feedback**
>
> Dear Reviewer HH2K,
>
> Thanks again for your insightful suggestions and comments. As the deadline for discussion is approaching, we are happy to provide any additional clarifications that you may need.
>
> In our previous response, we have carefully studied your comments and made detailed responses summarized below:
>
> 1. Clarified that learning the relevance between language and frames is based on the salient frames instead of the full video.
> 2. Clarified our two-stage sampling strategy and conducted additional experiments to show the superiority of our salient sampling strategy in terms of speed and memory.
> 3. Discussed how our SFP helped to ease the redundancy issue.
> 4. Provided the model capacity as well as the model performance of each method.
> 5. Explained how global and local alignments are used in the inference phase and why using global inference generates lower performance.
> 6. Clarified the complementarity of our salient frame sapling strategy and temporal infromation.
>
> We hope that the provided new experiments and additional explanations have convinced you of the merits of our submission.
>
> Please do not hesitate to contact us if there are other clarifications or experiments we can offer. Thanks!
>
> Thank you for your time!
>
> Best, \
> Authors

---

> > ### Comment · Reviewer_HH2K · 2022-08-07
> > **Rebuttal discussion**
> >
> > Thank you for the replies. I have some follow-up thoughts for some questions.
> >
> > **W2: [… Please see Section 3.4 or Figure 2 in the main paper for more details.]**
> >
> > Thank you for the clarification. However, I was trying to say, in Figure 2, only the multi-modal Fusion Transformer is trained with denoise data, and the vision and language encoders are still trained with noisy data. Given that 2 out of 3 sub-networks (and they are big and deep), I think the whole model still highly relies on the training on noisy data.
> >
> > But I actually found you have mentioned “obtains only 2–4 salient 14 frames per video for **cross-modal token-level alignment**” in the L13, so only using denoise data on the multi-modal Fusion Transformer makes sense to me now.
> >
> > **Q1: To clarify the two stages of our sampling strategy, we have added a schematic figure (i.e., Figure 1) in the supplementary material...
> > We have added these results in Table 1 of the supplementary material.**
> >
> > Thanks. The added figure helps a lot.
> >
> > **Q2: How does SFP help to ease the redundancy in the video? … loss might be necessary to address this problem).**
> >
> > > Most recent works utilize sparse sampling to sample 16 frames per video for video-language modeling. As we have mentioned in our response to Q1, our SFP further selects a few salient frames (e.g., 2 ones) out of 16. The use of much fewer frames by two-stage frame sampling indeed contributes to easing the redundancy in the video.
> >
> > I think reducing frames does not necessarily lead to reduce redundancy. Take Figure 1 as an example. There are two redundant frames out of 7 frames, so the redundancy rate is 28.6%. Assuming the SFP selects 4 salient frames, model may select the two salient frames (the second and the third frame) and the two redundant ones (because the two redundant frames both contain the witch and lady, and the similar score to the third frame might be high). Then the redundancy rate will increase to 2/4 = 50%.
> >
> > > In this work, the problem of sampling similar salient frames can almost be ignored.
> >
> > Does this mean redundancy is actually not an issue for the video-language tasks?
> >
> > **Q3: … We have added these results in Table 2 of the supplementary material.**
> >
> > Thank you for the table.
> >
> > **Q4: Thanks for pointing this out. In the inference phase, …,**
> >
> > Thank you for the clarification.
> >
> > **Q5:  …​​This indicates that the semantic correlation is as important as temporal information for most video tasks. Therefore, it is essential to exploit both semantic correlation and temporal information for video-language modeling…**
> >
> > Thank you for the answer!

---

> > > ### Author Response · Authors · 2022-08-07
> > > **Response to Rebuttal Discussion**
> > >
> > > Thanks for your detailed feedback. We are glad to see that our response address most of your concerns. For the newly-raised questions, our answers are as follows:
> > >
> > > **Q1. [However, I was trying to say, in Figure 2, only the multi-modal Fusion Transformer is trained with denoise data, and the vision and language encoders are still trained with noisy data. Given that 2 out of 3 sub-networks (and they are big and deep), I think the whole model still highly relies on the training on noisy data.]**
> > >
> > > **A:**  Yes, as videos are naturally noisy, the noisy issue in the training data is inevitable. The key idea of our LGDN is to handle the noisy problem when only noisy training data is available rather than to clean the original data at the very beginning. Therefore, we try to alleviate the negative effect of noise in the training process. Although our MVCL is still trained on noisy data (full video), we apply the frame-level MIL-contrastive loss and SFP mechanism to reduce the impact of noise as much as possible for MFCL and LSFM.  Note that the vision and language encoders are also trained with denoise data (i.e., filtered salient frames) in frame-level MIL-contrastive loss.
> > >
> > >
> > > **Q2. [I think reducing frames does not necessarily lead to reduce redundancy...Does this mean redundancy is actually not an issue for the video-language tasks?]**
> > >
> > > **A:** 1. On reducing frames and redundancy, there seems to exist some confusion. Our explanations are:
> > > * Take Figure 1 as an example (selected from MSRVTT). In our practice, we select $N_{salient} = 2$ salient frames out of 16 frames from this video and show only 7 frames in Figure 1 for conciseness. The relevance scores of Frame 1-7 in Figure 1 are [0.051, 0.356, 0.372, 0.341, 0.097, 0.082, 0.313], respectively. As we only select $N_{salient} = 2$ salient frames, the two redundant frames will not be selected in this example.
> > > * Meanwhile, as redundancy significantly affects **the speed and memory efficiency** (instead of performance), the redundancy rate should be computed over the entire video frames. Then assuming the SFP selects 4 salient frames out of 7 frames (consistent with the reviewer), the redundancy rate is still 2/7. Moreover, in our case (only 2 salient frames are selected), the redundancy rate is actually reduced from 2/7 to 0.
> > >
> > > 2. Since reducing the redundancy of video frames leads to high speed and memory efficiency in real-world applications, redundancy is still a key issue for video-language tasks.

---

> > > > ### Comment · Reviewer_HH2K · 2022-08-08
> > > > **Response to Rebuttal Discussion**
> > > >
> > > > **Q2**
> > > >
> > > > > Take Figure 1 as an example (selected from MSRVTT). In our practice, we select  salient frames out of 16 frames from this video and show only 7 frames in Figure 1 for conciseness. The relevance scores of Frame 1-7 in Figure 1 are [0.051, 0.356, 0.372, 0.341, 0.097, 0.082, 0.313], respectively. As we only select  salient frames, the two redundant frames will not be selected in this example.
> > > >
> > > > This is just an example. My intention was to try to show that reducing frames does not lead to reducing redundancy. As you can see, the scores are very high for those redundant frames, so it is very possible that the model selects one salient frame and one redundant frame for some cases, and the redundancy rate will still be 50% even when $N_{salient}=2$.
> > > >
> > > > > Meanwhile, as redundancy significantly affects the speed and memory efficiency (instead of performance), the redundancy rate should be computed over the entire video frames. Then assuming the SFP selects 4 salient frames out of 7 frames (consistent with the reviewer), the redundancy rate is still 2/7. Moreover, in our case (only 2 salient frames are selected), the redundancy rate is actually reduced from 2/7 to 0.
> > > >
> > > > Why should the redundancy rate be computed over the whole video? I thought the model only computed loss over those selected frames, so the redundant frames, which might hurt the training quality, might have more portion if fewer frames are selected.

---

> > > > > ### Author Response · Authors · 2022-08-08
> > > > > **Thanks a lot for your insightful discussion and looking forward to your post-rebuttal rating!**
> > > > >
> > > > > Thanks for your feedback again, which will help us further improve our final paper!
> > > > >
> > > > > **Q. [This is just an example. My intention was to try to show that reducing frames does not lead to reducing redundancy ... and the redundancy rate will still be 50%. I thought the model only computed loss over those selected frames, so the redundant frames, which might hurt the training quality, might have more portion if fewer frames are selected.]**
> > > > >
> > > > > **A:** Thanks for pointing it out. We agree with your definition of the redundancy rate. This "relative" redundancy rate may increase when selecting fewer frames in a simple video. To make it clearer, we reiterate our contributions:
> > > > > 1. removing ambiguous frames to achieve higher accuracy;
> > > > > 2. reducing some redundant frames for better efficiency.
> > > > >
> > > > > We appreciate that you recognize our first contribution about removing ambiguous frames. We clarify your concerns on redundant frames as follows.
> > > > > - Considering the salient frame scores are aggregated by mean pooling for the video-level prediction (see Line 199 in Section 3.5), redundant frames with similar scores may have a slight impact on the model performance. In other words, it will not hinder the performance.
> > > > > - Now we again take Figure 1 as an example, choosing frame-sets (2, 3), (2, 4), or (2, 3, 4) does not have much impact on the training of the model as the average scores of these different frame-sets are very similar. Thus an increase in the relative redundancy rate is acceptable.
> > > > > - However, the *de facto* paradigm utilizes all frames (including all relevant frames) in the training/inference stage, which significantly affects the speed and memory efficiency. As our SFP chooses a few salient frames with the highest relevance scores for MFCL/LSFM, we say that our SFP is capable to alleviate this problem (in other words, the "redundancy" issue).
> > > > > - We also agree that your suggested diversity loss may further help reduce redundancy in this case. We will update the discussions in our paper.
> > > > >
> > > > > Thanks for your time again! Please don’t hesitate to let us know if there are any additional clarifications we can offer, looking forward to your post-rebuttal rating!
> > > > >
> > > > > Best, \
> > > > > Authors

---

> > > > > > ### Comment · Reviewer_HH2K · 2022-08-09
> > > > > > **The final thought and the new rating**
> > > > > >
> > > > > > Thanks for your reply.
> > > > > >
> > > > > > I still have some concerns about this redundancy issue, such as (1) choosing the highest relevance score does not equal easing the redundancy issue, and (2) subsampling or mean pooling is not the main contribution of this work even though they can ease redundancy. I think the diversity loss can address my question, so look forward to your results on this. Even though I still have the question, the current form of this work is worth being accepted in my opinion after most of my concerns are addressed. Therefore, I increase the soundness to 3, presentation to 3 (somewhere between 2 and 3), and the overall score to 6.
> > > > > >
> > > > > > Thanks again for your time to discuss my questions.

---

> > > > > > > ### Author Response · Authors · 2022-08-09
> > > > > > > **Thanks again for spending a huge amount of time on our paper!**
> > > > > > >
> > > > > > > Dear Reviewer HH2K:
> > > > > > >
> > > > > > > Thanks again for spending a huge amount of time on our paper, which has helped us improve the quality and clarity of the paper! We are glad to see that our response has addressed most of your concerns. We will carefully revise our paper according to your suggestions.
> > > > > > >
> > > > > > > Thanks for your time and efforts again!
> > > > > > >
> > > > > > > Best, \
> > > > > > > Authors

---

### Official Review · Reviewer_kV1h · 2022-07-16

**Rating:** 8
**Confidence:** 5
**Soundness:** 4 excellent
**Presentation:** 4 excellent
**Contribution:** 3 good

**Summary:**

This paper tackles one important issue in video-language modeling, that is,  previous works made assumptions that video frames and the text descriptions for the video are semantically correlated. However, this assumption is not valid for the real world <video,text> pairs, since first,  the video-level text descriptions may not cover all information in the video, and second, a raw video often contains noisy or meaningless information that does not appear in the text descriptions. The previous correlation assumption did not take account of these noises, and the self-attention mechanism still leaves mis-aligned frames which harm cross-modal alignment.

This paper proposed an E2E language-guided denoising network, LGDN, for video-language modeling. LGDN includes (1) a salient frame proposal (SFP) mechanism to dynamically filter out irrelevant or redundant frames and sparsely sample salient frames to improve video-language modeling,   (2) cross-modal interactions at three levels, i.e., salient frame matching at the token-level guided by language, momentum frame-level MSL-contrastive learning (MFCL), and momentum video-level contrastive learning (MVCL). Experimental results showed that LGDN outperforms the current SOTA. Ablation studies further demonstrated the importance of tackling the noise issue.


**Questions:**

The paper is clear written. However, the authors mentioned the pre-training is set up as described in the paper due to restricted computation resources. It would be useful to explain the experimental setup alternative if more computation resources are available, that is, how to scale up the pre-training setup.

**Limitations:**

The paper missed the important section of Limitations (and potential negative societal impacts). It is highly desired that the authors add discussions on limitations and potential negative societal impacts of this work.

**Strengths And Weaknesses:**

Strengths:

1. This paper addressed an important limitation in previous works of video-language modeling, that is, the impractical assumption that video frames and the text descriptions for the video are semantically correlated. There is no sufficient tackling of noises in raw videos. The existing employment of self-attention mechanism cannot fully address this problem. These irrelevant or redundant information still harms cross-modal alignment.  Hence, this work is valuable to the research community.

2. The SFP mechanism and the three-level cross-modal interactions are sound, well motivated, and have good novelty.  SFP is achieved through exploring MVCL and MFCL. After obtaining the language-guided salient frames, LGDN also has a language-guided salient frame matching (LSFM) module to conduct token-level semantic alignment between visual patches and words for improved performance. The final loss of LGDN is the combined loss of MVCL, MFCL, and LSFM.

3. The Related Work section is clearly written. It summarizes significant works in the past and their limitations, and clearly explained the choice of momentum frame-level MSL-contrastive learning to help address the mis-aligned frames.

4. The paper also conduct the interesting investigations of the four strategies for estimating relevance scores.

5. The evaluations are comprehensive, on four public text-video retrieval datasets and one VQA dataset. The evaluation settings support a fair comparison to previous models.  Performance gains from LGDN over existing approaches, including SOTA, are quite strong.  Experimental results showed that LGDN outperforms previous methods, including the current SOTA,  by a large margin on text<->video retrieval on MSR-VTT. LGDN also outperforms methods exploiting extra modalities or those pre-trained on very large video data.  LGDN also shows promising model capacity, as its performance is significantly improved when trained on much larger pre-training data. The extensive evaluations on other text-video retrieval datasets and VQA also demonstrated that LGDN outperforms existing approaches including the current SOTA.

6. Ablation studies verified the contributions from the proposed SFP, MVCL, MFCL, and LSFM. Last but not least, the visualization results are interesting and helpful for directly illustrating the importance of addressing the noisy frame issue and showing that the SFP mechanism helps LGDN to improve video-language modeling.

7. Although the code is not released, the main body and Appendix provide enough details to help reproducibility. The Appendix also includes more detailed experimental results on analyzing effect of SFP mechanism, different relevance score estimations, and memory bank sizes, and useful additional visualization results.

Weaknesses:

1. The paper missed the important section of Limitations (and potential negative societal impacts). It is highly desired that the authors add discussions on limitations and potential negative societal impacts of this work.

---

> ### Author Response · Authors · 2022-08-01
> **Response to Reviewer kV1h**
>
> Thank you for the positive comments and insightful suggestions.
>
> **Q1. [Lack of limitations and potential negative societal impacts.]** \
> **A:** Thanks. We have added them in Section 1 of the supplementary material. We also show it below.
> - **Limitations.** The key idea of our LGDN is to propose the SFP mechanism to filter out noisy/redundant frames for fine-grained semantic alignment, along with MVCL for capturing global temporal information. In most downstream tasks, these two modules are complementary to each other. And we also observe that only a few salient frames (e.g., 2 ones) are enough for most downstream tasks, and thus we do not consider aggregating temporal information across salient frames.
> However, the SFP mechanism may need to be slightly changed when facing specified scenarios (e.g., long-term complicated videos over 30 minutes that highly rely on temporal information).
> On the one hand, we could adjust the weights between the two modules (MVCL and SFP) according to the situation. On the other hand, we can split the full video into several clips (e.g., 3 minutes per clip), apply our SFP mechanism on each clip, and obtain the salient frames from all clips. In this way, we could consider aggregating temporal information across salient frames.
> - **Potential Negative Societal Impacts.** Video-language learning, especially large-scale video-language modeling, has developed rapidly over the past few years and led to the greatest advance in search engines, video recommendation, and multimedia data management. Despite its effectiveness, existing video-language pre-training models still face possible risks. As these models often rely on a large amount of web data, they may acquire biases or prejudices (especially in search engines and recommendation systems), which must be properly addressed before model deploying.
>
> **Q2. [Explain the pre-training setup alternative if more computation resources are available.]** \
> **A:** Good question! If more computation resources are available, the pre-training setup alternative could be:
> - Larger pre-training datasets (e.g., HowTo100M and LAION-400M [1]) are utilized to obtain better performance.
> - A larger encoder (e.g., ViT-H) is used as our backbone.
> - The larger batch size and more modality data (e.g., video, image, and motion) are also good choices, which are beneficial for video-language learning.
>
> [1] Christoph Schuhmann, et al. "LAION-400M: Open Dataset of CLIP-Filtered 400 Million Image-Text Pairs." NeurIPS Workshop 2021.
>
> Thanks for your time and effort again! For any other questions, please feel free to let us know during the rebuttal window.

---

> > ### Comment · Reviewer_kV1h · 2022-08-10
> > **Response to authors' response**
> >
> > I would like to thank the authors for addressing all of my comments. And, the added limitations and potential negative societal impacts are reasonable to me. Thank you!

---

### Author Response · Authors · 2022-08-01
**General Response: Contributions and New Experiments**

We sincerely appreciate all reviewers’ time and efforts in reviewing our paper. We are glad to find that reviewers generally recognized our contributions:

* **Model.** Introducing a new way and cool idea to address the noisy problem in video [kV1h, HH2K, UdJY]; the proposed salient frame proposal mechanism is simple and effective [kV1h, HH2K, UdJY]; the impact can be extended to other domains [HH2K].
* **Experiment.** Extensive experiments are performed on video-text datasets and the experimental results are promising [kV1h, HH2K, UdJY].
* **Writing.** The paper is well organized and easy to read [kV1h, UdJY]; the Related Work section is clearly written [kV1h].

And we also thank all reviewers for their insightful and constructive suggestions, which help a lot in further improving our paper. In addition to the pointwise responses below, we summarize supporting experiments added in the rebuttal according to reviewers’ suggestions.

**New Experiments**

* The speed and memory cost of different sampling strategies [HH2K].
* Comparision among different methods in terms of the model capacity [HH2K].
* More results by applying our SFP mechanism to other frame sampling techniques [UdJY].

We hope that our pointwise responses below could clarify all reviewers’ confusion and alleviate all of their concerns. We'd like to thank all reviewers’ time again.

---

### Author Response · Authors · 2022-08-06
**Looking forward to your post-rebuttal feedback**

Dear ACs and reviewers:


Thanks again for all of your constructive suggestions, which have helped us improve the quality and clarity of the paper!


The discussion period will end soon in 4 days. We appreciate it if you take the time to read our rebuttal and give us some feedback. Please don’t hesitate to let us know if there are any additional clarifications or experiments that we can offer, as we would love to convince you of the merits of the paper. We appreciate your suggestions.


Thanks for your time and efforts.


Best, \
Authors

---

### Meta-Review · Area_Chair_fF9N · 2022-08-28

**Recommendation:** Accept
**Confidence:** Certain

**Metareview:**

This paper proposes Language-Guided Denoising Network (LGDN) with the goal of addressing the redundancy and noisy alignment issues in video-language modeling, by dynamically filtering out misaligned or redundant frames under the language supervision and obtains only 2-4 salient frames per video for cross-modal token-level alignment. Experiments on multiple benchmarks obtain good results. Reviewers appreciated the intuitive salient frame proposal mechanism and well-written paper but also had several questions on more ablations and efficiency details and fair comparisons, some of which were answered by the authors but some remaining were left unanswered for some reviewers and they were promised in the final version (e.g., diversity loss, SOTA methods with CC12M, etc.) so it will be good to add these.


**Award:**

No

---

### Decision · Program_Chairs · 2022-09-14

Accept